# Identifying T-cell clubs by embracing the local harmony between TCR and gene expressions

Yiping Zou [ID] [1,2,4], Jiaqi Luo [ID] [1,2,4], Lingxi Chen [ID] [1,2], Xueying Wang [1,2,3], Wei Liu [1,2], Ruo Han Wang [1,2] & Shuai Cheng Li [ID] [1,2 ✉]

## Abstract

T cell receptors (TCR) and gene expression provide two complementary and essential aspects in T cell understanding, yet their diversity presents challenges in integrative analysis. We introduce TCRclub, a novel method integrating single-cell RNA sequencing data and single-cell TCR sequencing data using local harmony to identify functionally similar T cell groups, termed 'clubs'. We applied TCRclub to 298,106 T cells across seven datasets encompassing various diseases. First, TCRclub outperforms the state-of-the-art methods in clustering T cells on a dataset with over 400 verified peptide-major histocompatibility complex categories. Second, TCRclub reveals a transition from activated to exhausted T cells in cholangiocarcinoma patients. Third, TCRclub discovered the pathways that could intervene in response to anti-PD-1 therapy for patients with basal cell carcinoma by analyzing the pre-treatment and post-treatment samples. Furthermore, TCRclub unveiled different T-cell responses and gene patterns at different severity levels in patients with COVID-19. Hence, TCRclub aids in developing more effective immunotherapeutic strategies for cancer and infectious diseases.

**Keywords** Integration; Local Harmony; Single-Cell Analysis; T-Cell Clustering
**Subject Categories** Computational Biology; Immunology

## Introduction

The immune system is an intricate network comprising cells, tissues, and organs that safeguard the body from infections and illnesses. T cells are crucial in recognizing and eliminating infected or cancerous cells among the various components of the immune system. The diversity of T cells is reflected in the diversity of their antigen receptors, known as the T cell receptor (TCR). TCR allows T cells to recognize pathogens and antigens, making them essential for effective immune responses against pathogens and cancer cells

(De Simone et al, 2018; Emerson et al, 2017; Rocamora-Reverte et al, 2021).

One of the primary challenges in immunology is identifying functionally similar T cells. Several computational methods have been developed to cluster T cells to address this challenge based on the similarity of the complementarity-determining region 3 (CDR3) sequences. CDR3 is a highly variable region of TCR and is directly involved in the binding of epitopes presented by major histocompatibility complex (MHC) molecules (Stadinski et al, 2014; Chronister et al, 2021a). TCRdist (Dash et al, 2017; Mayer-Blackwell et al, 2021) applies a pairwise distance metric to quantify the similarity of CDR3 sequences. GLIPH (Glanville et al, 2017) focuses on identifying TCRs targeting specific antigens, grouping TCR sequences according to their shared amino acid motifs and global sequence similarity. GLIPH2 (Huang et al, 2020) is an improved version of GLIPH that employs Fisher's exact test to identify k-mer motifs and to revise global similarity to accelerate the processing of large datasets. ClusTCR (Valkiers et al, 2021) builds similarity graphs for every supercluster divided by Facebook artificial intelligence similarity search (Faiss) and then uses Markov clustering algorithm (MCL) to identify TCR clusters within each graph. GIANA (Zhang et al, 2021a) uses a geometric isometry-based method to transform CDR3 sequences into numerical vectors that preserve their similarity and then uses a classic nearest-neighbor search to group TCRs by their Euclidean distances.

Although some methods, such as TCRdist (Dash et al, 2017; Mayer-Blackwell et al, 2021) and GIANA (Zhang et al, 2021a), employ a nearest-neighbor strategy to identify clusters directly based on TCR distances, the clusters may be unreliable in accurately reflecting the diversity of functions due to the neglect of the underlying T-cell hierarchy (Wu and Wu, 2020). In a recent study (Chen and Li, 2022), we have shown that cell-cell relationships can be mapped to a hierarchical structure of multiple layers, where cellular homogeneity increases from the root to the leaves. In the case of T cells, the root layer represents the entire population, followed by the subpopulation layer that contains a broad category of T cells, such as helper T cells and naive T cells. The club layer, which is the next layer in the hierarchy, provides a finer resolution of cellular homogeneity by grouping individual T cells based on their similar functions. The hierarchy is built in a specific order, sequentially adding each layer to the internal cell hierarchy.

[1]Department of Computer Science, City University of Hong Kong, Hong Kong, China. [2]Department of Computer Science, City University of Hong Kong Shenzhen Research Institute, Shenzhen, China. [3]Department of Computer Science, City University of Hong Kong (Dongguan), Dongguan, China. [4]These authors contributed equally: Yiping Zou, Jiaqi Luo. ✉E-mail: shuaicli@cityu.edu.hk

By contrast, methods that build clusters by the nearest-neighbor search are limited to basic layer information and generate flat cell groups, failing to fully capture the functional diversity of the cells.

Moreover, while TCR sequences alone are crucial for T cell identification, they do not fully encompass the functional diversity of T cells, since functional diversity is also influenced by their gene expression profiles (Cheadle et al, 2005; Chtanova et al, 2005). Integrating TCR sequences with single-cell RNA sequencing (scRNA-seq) data can provide a more comprehensive view of the T cell landscape and reveal the phenotypes and functions of different T cell subsets. TESSA (Zhang et al, 2021a) leverages both gene expression and TCR sequences. It uses a Dirichlet process to map the TCR repertoire's functional landscape and identify antigen-specific TCR clusters. However, one limitation of TESSA is that it assumes a fixed prior distribution for the Dirichlet process, which may not reflect the true diversity and complexity of the TCR repertoire. Additionally, TESSA requires all T cells in every iteration of the calculation to infer the posterior distribution of the latent variables and parameters. It may not be suitable for analyzing highly dynamic T-cell populations. Another clustering method, CoNGA (Schattgen et al, 2022), also utilizes both gene expression and TCR sequences through statistical analysis of gene expression and TCR similarity graphs. While CoNGA considers the neighbors of T cells, it constructs separate graphs for gene expression and TCR sequences and then compares them, rather than integrate them into a single graph.

Here, we propose TCRclub, a novel approach that identifies the functional relevance of T cells. TCRclub receives single-cell gene expressions and numeric embeddings of TCRs as input. It aims to bridge the gap between gene expression and TCRs by modeling the relationship between pairwise TCR embedding and pairwise expression distances according to local harmony. Local harmony refers to the homogeneity of local neighbors surrounding an instance (e.g., a cell), since the neighbors in the distance space are more likely to have similar characteristics and belong to the same category (Uddin et al, 2022; Cover and Hart, 1967). By emphasizing local harmony, TCRclub reduces noise and increases the robustness of integration. Considering the built-in cell structure, TCRclub builds the T-cell hierarchy based on the residual-distance matrix obtained by integration and identifies the T-cell clubs. Finally, the TCRclub is repeated multiple times to obtain the consensus results of the clubs as the final output.

We applied TCRclub to 298,106 T cells from seven scRNA-seq + scTCR-seq datasets encompassing various samples representing distinct diseases. First, we validated that our model performed better than existing methods in effectively grouping T cells into peptide-major histocompatibility complex (pMHC)—specificity clubs, with a higher clustering purity of 52.25% and clustering coverage of 87.45% across over 400 pMHC categories. Furthermore, we showcased the ability of TCRclub to unveil the functional landscape of T cells in diverse biological contexts, including cholangiocarcinoma, Anti-PD1 therapy, and SARS-CoV-2 infection.

# Results

## TCRclub identifies T-cell clubs by TCR sequences and gene expressions

TCRclub takes paired gene expressions and TCR sequences as input to identify the T-cell clubs. A T-cell club is defined as a

subpopulation of T cells that exhibits high functional similarity within the hierarchy of the T cells (Chen and Li, 2022).

Given that CDR3$\beta$ is the most diverse and variable component of TCR (Rosati et al, 2017), our study specifically focuses on the CDR3$\beta$ sequence as the input for TCRclub. The term "TCR embedding" in our study refers to the numerical vector obtained by encoding the CDR3$\beta$ sequence using an encoder-classifier approach (Luo et al, 2023). This CDR3$\beta$ embedding represents the TCR for subsequent integration with gene expression data. A T-cell clone is defined as a group of T cells sharing identical CDR3$\beta$ sequences from the same sample (Zhang et al, 2021a). The goal of TCRclub is to group functionally similar T cells into a club. Since T cells within a clone are likely to have the same antigen-specificity (Smith et al, 2021; Chronister et al, 2021) and tend to have similar gene expressions (Schattgen et al, 2022) (Appendix Fig. S1), TCRclub takes the T-cell clone as the basic unit.

As shown in Fig. 1A, TCRclub takes paired gene expressions and TCR embeddings as two input matrices. TCRclub consists of two main steps: (1) iterative integration of gene expressions and TCR embeddings using local harmony to produce a residual-distance matrix at convergence and (2) hierarchical clustering of T-cell clones based on the residual-distance matrix to identify T-cell clubs. TCRclub repeats this workflow for a specified number of rounds and obtains the consensus result by aggregating the T-cell clubs of rounds (Fig. 1B).

## TCRclub discovers latent representations of TCRs

During the integration of scRNA and TCR data, TCRclub assigns varying levels of importance to different dimensions of TCR embeddings, thereby highlighting regions of the TCR sequence potentially critical for antigen recognition and function. We used the concept score metric (Brocki and Chung, 2019) to evaluate the TCRs clustered by TCRclub. The concept score, obtained by calculating the dot product between the TCR embedding and the generated concept vector, allowed us to interpret the latent representation of high-level concepts within the TCR sequence (Brocki and Chung, 2019). Using concept scores, we generated density histograms for each club and compared them with other clubs within the same sample (Fig. 2A,B; Appendix Fig. S2). Remarkably, when each club corresponds to prominent epitopes, a distinct separation between the clubs based on their peak positions was observed (Fig. 2A). In contrast, when the club does not correlate with prominent epitopes, a relatively closer separation between the clubs was noted (Fig. 2B). Moreover, for the clubs associated with prominent epitopes (Fig. 2A), we generated saliency maps using Deepexplain (Ancona et al, 2017) for the TCRs with the highest concept scores within each club. These maps unveiled shared attention positions among the TCRs within the same club (Fig. 2C; Appendix Fig. S2). In summary, our findings underscore that TCRclub effectively captures both similarities and distinctions within and among TCR clubs.

## TCRclub demonstrates better performance in clustering T cells into pMHC-specificity clusters

In this section, we assess the ability of TCRclub to cluster T cells into various pMHC-specificity groups. We utilized a dataset (Data ref: Francis et al, 2021) containing 55 samples of single-cell gene

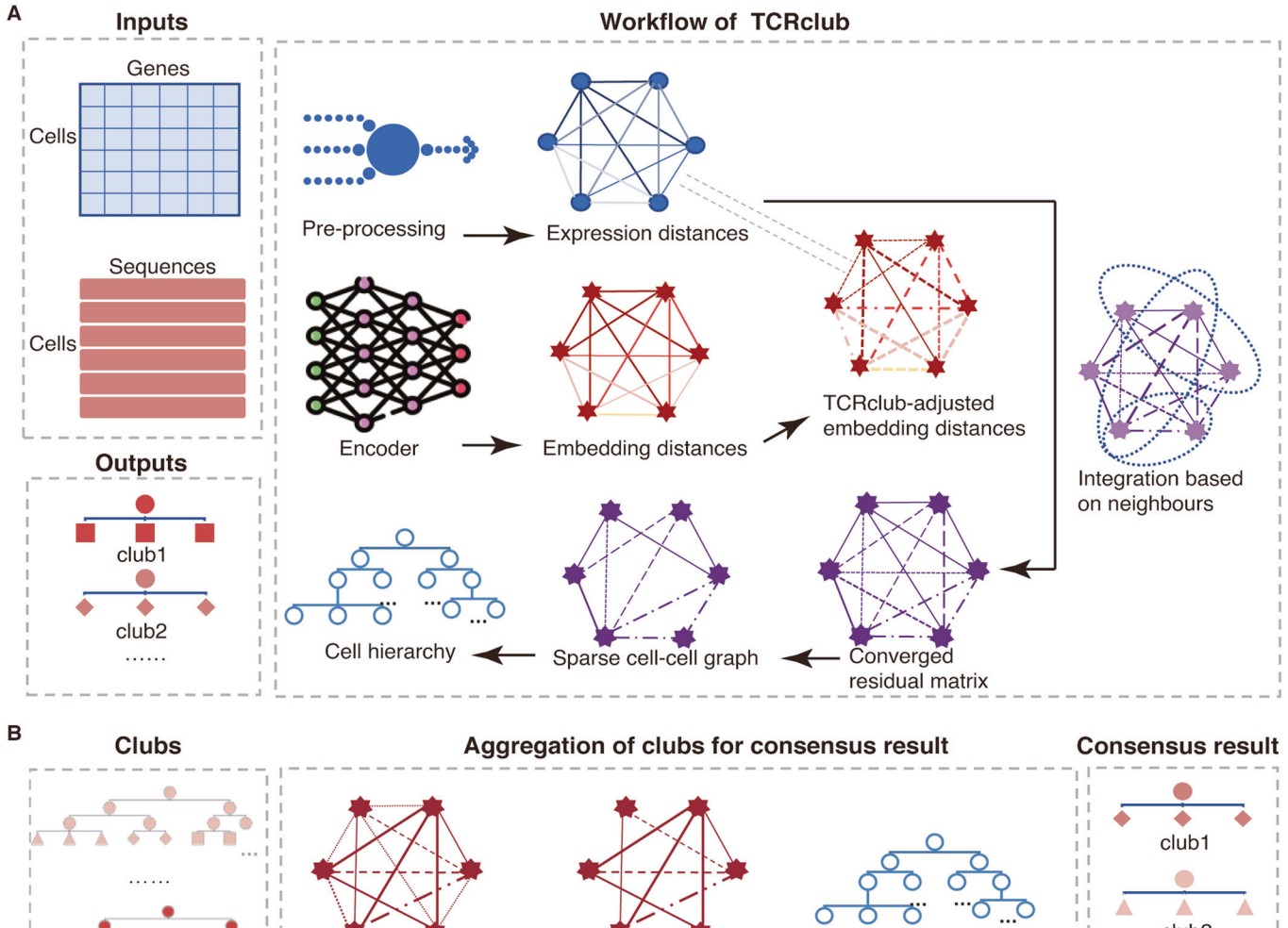

**Figure 1. Schematic overview of TCRclub.**

(A) The workflow of TCRclub. (B) Aggregation of clubs for consensus result.

expression coupled to TCR profiling, with pMHC specificity verified for each T cell by tetramer staining technique. Given the large number of pMHC categories (over 400), accurately distinguishing between functionally similar T-cell clusters posed a challenge. To benchmark TCRclub, we compared its performance with TESSA, CoNGA, GIANA, ClusTCR, and the series of GLIPH using the same samples and their default parameter settings, while quantifying the clustering performance. The clustering accuracy was assessed through clustering purity which counts the number of T-cell clones targeting the most frequent pMHC in each T-cell clone cluster, while clustering coverage represents the percentage of T-cell clones that can be assigned to clusters among all the input T-cell clones. In most scenarios, tuning the clustering to improve purity typically reduces coverage, and vice versa. Therefore, we introduce clustering effectiveness, defined as the product of the clustering purity and coverage, to capture the trade-off between these two critical aspects of T-cell clustering performance.

We selected T cells with single-specificity (i.e., not considering multi-matching T cells) to provide an intuitive illustration of the result of its cluster. TCRclub exhibited a higher level of agreement with the ground truth, as demonstrated by the colormap (Fig. 2D). The clustering results obtained using TCRclub constructed a tree-like structure for T cells (Fig. 2E), where the level of functional similarity increases as one moves from the root towards the leaves (Chen and Li, 2022) (Appendix Fig. S3). The "club layer", the focus of our study, represents the finest resolution of functional similarity, and it is positioned just before the last layer (the individual T cells). Since each cell has a unique TCR, the last layer can also be regarded as T-cell clones. In contrast, other methods resulted in flat cell groups (Fig. 2F–H). We used the bar chart to display the purity of the T-cell clone clusters (Fig. 2E–H). We observed that TCRclub demonstrated higher purity and coverage compared to other methods.

We compared TCRclub and other methods across all samples. TCRclub achieves a higher average clustering purity of 52.25% and

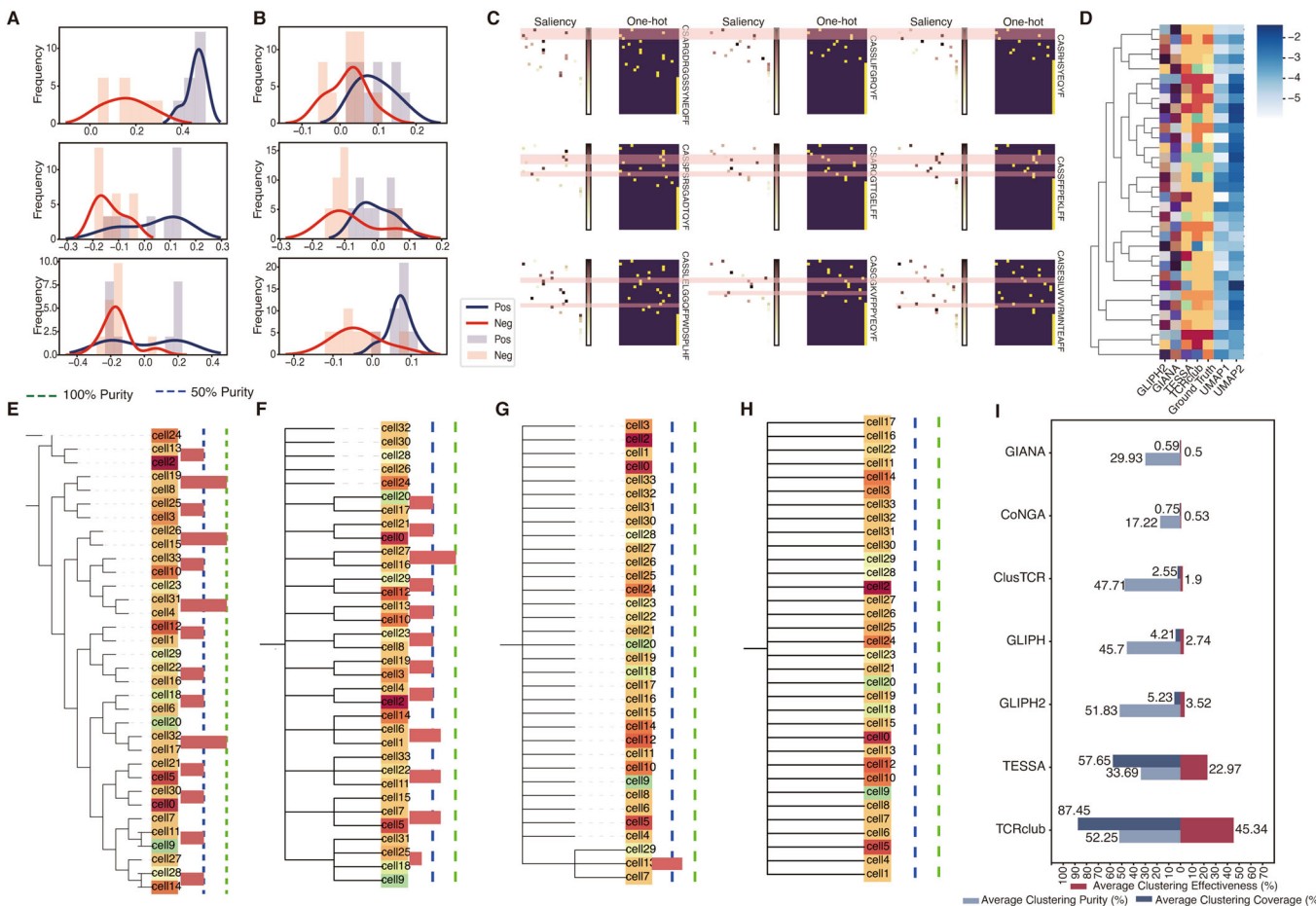

**Figure 2. Performance evaluation of TCRclub for T cell clustering.**

(A–C) Density histograms of three clubs obtained from the same sample illustrate the distribution of concept scores. The blue histograms represent the club of interest, while the red histograms represent other clubs from the same sample. (A) Three clubs presenting different prominent epitopes. (B) Three clubs without prominent epitopes. (C) Saliency maps for the three clubs in (A). Each row displays the saliency maps for the top four TCRs with the highest concept scores within the club. The dark-background heatmap indicates the one-hot embedding (see Appendix Supplementary Methods) for each TCR, while the white-background heatmap represents the saliency of each amino acid in the TCR. The color bar denotes the saliency level, with deeper colors indicating higher importance. Shared attentional positions are highlighted in red. (D) Heatmap comparing the clustering results of different models with the ground truth. RNA expressions were dimensionally reduced by uniform manifold approximation and projection (UMAP) and are depicted in blue. The ground truth shows the verified pMHCs of T cells, with each pMHC assigned a unique color. In the ground truth, the same colors depict T cells with the same pMHC. For the models, T cells were colored according to the clustering result. For example, if the largest number of T cells in the cluster is against a particular pMHC, then all the T cells in this cluster will be assigned the color corresponding to that pMHC in the ground truth colormap. If a T cell does not belong to any cluster, it is assigned a color that is not present in the ground truth colormap. Therefore, the clustering result that aligns more closely with the colormap of the ground truth indicates higher purity and clustering coverage. (E–H) iTOL plots display the hierarchy of the clustering results from TCRclub (E), TESSA (F), and GLIPH2 (G). GIANA, CoNGA, and ClusTCR did not generate any clusters in the depicted example (H). Same as in (D), T cells are color-coded based on the verified pMHCs, where identical colors represent T cells recognizing the same pMHC. TCRclub (E) exhibits a tree-like structure, where T cells associated with the same node in the club layer are anticipated to share similar functions. Conversely, alternative models. Other models (F–H) generate produced flat cell groupings, where T cells affiliated with the same child node are expected to exhibit similar functions. The blue and green dotted lines mark the 50 and 100% clustering purity, respectively. (I) Comparison of TCRclub with other models in terms of average effectiveness, purity, and coverage of the dataset.

a coverage of 87.45% (Fig. 2I). When considering both clustering purity and coverage, TCRclub achieved a significantly higher clustering effectiveness of 45.34%, indicating its balanced ability to detect functionally similar clusters across various samples. In contrast, other methods demonstrated lower clustering effectiveness, regardless of whether they used only TCRs (e.g., GLIPH series and GIANA) or both TCRs and RNA (e.g., TESSA and CoNGA). Most methods had clustering effectiveness below 10%. The most comparable method, TESSA, achieved only 22.97%, significantly lower than TCRclub, indicating their inability to balance clustering

purity and coverage effectively. The methods compared frequently produce no T-cell cluster in certain samples, highlighting their stringent dataset requirements and limited generalizability.

We further evaluated conditions that might affect TCRclub's clustering performance. Because of the prevalence of dropouts in scRNA data, we examined whether these dropouts would affect TCRclub's clustering performance or not. By introducing additional dropouts into the raw gene expression counts while maintaining the same preprocessing steps, we found that increased dropouts could negatively impact both average clustering purity

and clustering coverage (Appendix Fig. S4). We also investigated how extreme clonal expansion might influence clustering outcomes. In simulations of extreme clonal expansion (see Appendix Supplementary Methods), we observed that this condition does not affect clustering results compared to scenarios without such expansion (Appendix Fig. S5).

## TCRclub reveals the activated-to-exhausted transition in cholangiocarcinoma patients

In this investigation, employing TCRclub, we explored the dynamic trajectory of T cells within cholangiocarcinoma (CCA) patients. Specifically, we utilized TCRclub to analyze a dataset consisting of matched scRNA-seq and scTCR-seq profiles of T cells derived from five patients diagnosed with cholangiocarcinoma (Shi et al, 2022b; Data ref: Shi et al, 2022a). This dataset includes samples obtained from peripheral blood (PB), the tumor's primary site, and lymph nodes. First, we obtained the regression coefficients between gene expression distances and TCR distances for each clone using TCRclub (see Methods). We then compared these coefficients across PB, tumor, and lymph node tissues. Interestingly, we found no significant differences in the regression coefficients between the different tissues (Appendix Fig. S6). This consistency suggests that the relationship between gene expression and TCR repertoire remains stable across different tissue origins in CCA patients. We annotated the cells of this dataset based on the annotation guidelines outlined by Shi et al (Shi et al, 2022b) (Appendix Figs. S7–S9). We integrated PB and tumor samples, or lymph node and tumor samples (if available), for each patient and subsequently applied TCRclub to the integrated samples. For a better joint analysis of different samples, we recommend integrating T-cell gene expressions and removing batch effects before applying TCRclub to cells from different samples. As an example, we compared integrated PB-tumor samples with unintegrated PB-tumor samples from five patients as an instance. We applied TCRclub to the groups, respectively. We calculated the ratio of clubs containing T-cell clones from both PB and tumor (i.e., mixed clubs). The results showed a significantly higher ratio of mixed clubs in the integrated samples compared to the unintegrated ones (Appendix Fig. S10). This finding suggests that integrating gene expression data is crucial for the joint analysis of different samples; otherwise, TCRclub tends to cluster T cells with those from the same sample.

For the T cells sharing identical CDR3$\beta$ sequences within the same sample, we defined them as a T-cell clone, and our observations indicated that cells within a given clone tend to exhibit similar gene expressions (Appendix Fig. S1). However, T cells with the same CDR3$\beta$ sequences from different samples should not be regarded as a single clone, as their gene expression could be very different due to variations in the microenvironment. We integrated different samples from the same patient across various diseases to assess whether the similarity in gene expression among T cells sharing the same CDR3$\beta$ sequences differed between cells from the same sample and those from different samples (Appendix Fig. S11). Our analysis revealed a significant difference in gene expression similarity, suggesting that T cells with identical CDR3$\beta$ sequences exhibit distinct gene expression profiles across samples, even after removing batch effects. Therefore, in our work, when applying TCRclub to T cells from different samples, gene expressions were integrated prior to TCRclub's analysis. Clones sharing identical CDR3$\beta$ sequences from different samples were treated as distinct clones, ensuring that all T-cell clones from each sample could be analysed.

After applying TCRclub to the integrated PB (or lymph) and tumor samples, we categorized PB (or lymph)-derived T cells as 'tumor-unrelated cells' if their club comprised solely PB (or lymph) T cells. Conversely, PB (or lymph)-derived T cells were labeled as 'tumor-related cells' if their club included both PB (or lymph) T cells and tumor T cells. In lymph-derived T cells from three patients (Fig. 3A), upon comparing the numbers of tumor-related and tumor-unrelated T cells in each T-cell subtype, we observed a substantial number of tumor-related activated T cells (Fig. 3B; Appendix Fig. S12). Similarly, in PB-derived T cells from five patients (Fig. 3C), we noted a significant presence of activated T cells associated with tumor T cells in each subtype (Fig. 3D; Appendix Fig. S12). The findings validate the successful migration of activated T cells from lymph nodes or PB and infiltrate into the tumor microenvironment (TME) in response to tumor-specific antigens. Within the tumor (Fig. 3E), a cell was termed 'PB-related T cells' if its club contained PB T cells; otherwise, it was classified as 'PB-unrelated T cells.' We found a significant number of activated cells, such as effector and memory CD8 cells, were associated with PB, whereas many exhausted CD8 T cells in the tumor were unrelated to PB T cells (Fig. 3F; Appendix Fig. S12). The presence of activated effector and memory CD8 T cells associated with PB in tumors indicates immune infiltration from systemic circulation into the TME. The dissociation between exhausted CD8 T cells and PB T cells in the tumor suggests that the TME may employ immune evasion mechanisms to evade systemic immune surveillance after the activated T cells infiltrated into TME.

Given the critical role of CD8 T cells in mounting an immune response against tumor cells, primarily through their capacity to directly eliminate tumor cells, we isolated all CD8 T cells within the tumor primary locus for subsequent trajectory analysis (Appendix Fig. S13A) and pseudotime analysis (Appendix Fig. S13B). Our observations revealed progressive exhaustion of effector and memory CD8 T cells over time, while a subset of memory and effector CD8 T cells persisted. Our analysis of CD8 T-cell distribution within the clubs echoed the findings: a majority of the clubs exhibited a simultaneous presence of effector, memory, and exhausted CD8 T-cell types, whereas a smaller subset of clubs only displayed the presence of effector and memory CD8 T cells (Appendix Fig. S14). Specifically, we investigated the differentiation dynamics between PB-related activated T cells and PB-unrelated activated T cells. We applied trajectory and pseudotime analysis to effector CD8 T cells (Fig. 3G,H). We identified that PB-related effector CD8 T cells predominantly occupied the early stages, gradually transitioning into PB-unrelated effector CD8 T cells, and ultimately culminating in exhaustion (Fig. 3I,J). A similar trend was observed in PB-related memory CD8 T cells, where their numbers decreased over time. Some PB-related memory CD8 T cells transitioned into PB-unrelated memory CD8 T cells and persisted in the tumor, while others progressively progressed toward exhaustion (Appendix Fig. S15).

To elucidate the underlying mechanisms behind the transition from PB-related T cells to PB-unrelated T cells, we conducted a differential gene expression (DEG) analysis on effector (Fig. 3K) and memory CD8 T cell subsets (Fig. 3L) to uncover the

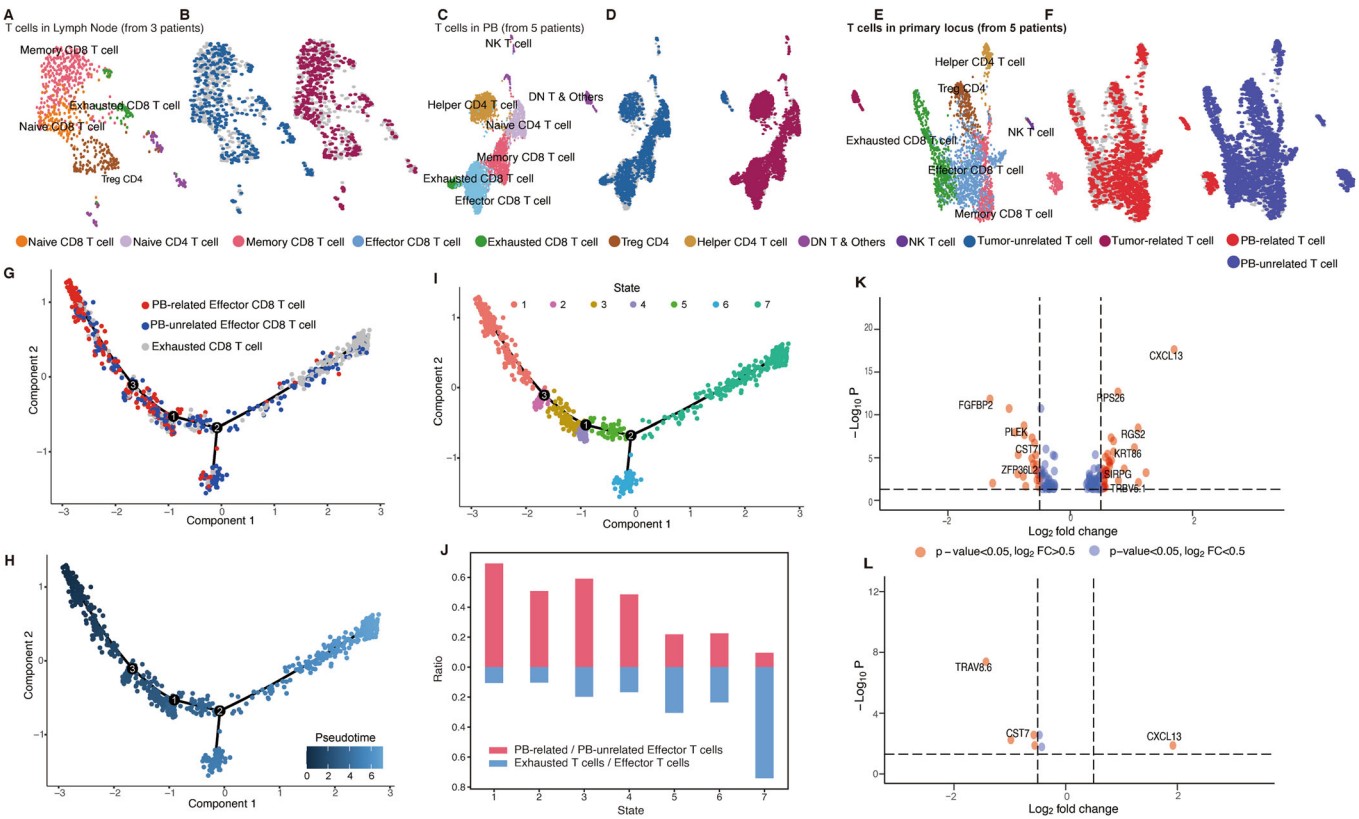

**Figure 3. TCRclub reveals the dynamic journey: early-stage activated T cells transition to late-stage exhausted phenotype in cholangiocarcinoma patients.**

(A) UMAP plot of lymph T cells labeled with distinct colors and accompanied by cell type annotations. (B) Tumor-unrelated T cells (left) and tumor-related T cells (right) depicted in lymph samples. (C) UMAP plot of PB T cells labeled with distinct colors and accompanied by cell type annotations. (D) Tumor-unrelated T cells (left) and Tumor-related T cells (right) visualized in PB samples. (E) UMAP plot showing T cells labeled with varying colors for tumor primary locus samples, with corresponding cell type annotations. (F) PB-related T cells (left) and PB-unrelated T cells (right) are depicted in tumor samples. (G) Trajectory plot depicting the progression of effector CD8 T cells and exhausted CD8 T cells within tumor samples. (H) Pseudotime trajectory plot of effector CD8 T cells and exhausted CD8 T cells within tumor samples. (I) Trajectory plot demonstrating the states of effector CD8 T cells and exhausted CD8 T cells within tumor samples. (J) The ratio of PB-related to PB-unrelated effector CD8 T cells, along with the ratio of effector CD8 T cells to exhausted T cells in each state depicted in (I). Monocle 2 was employed for trajectory and pseudotime analysis. (K) Volcano plot displaying DEGs for PB-unrelated effector CD8 T cells ($n = 520$) compared with PB-related effector CD8 T cells ($n = 440$). The $p$ values were generated from the two-sided Wilcoxon rank-sum test. (L) Volcano plot depicting DEGs for PB-unrelated memory CD8 T cells ($n = 313$) compared with PB-related memory CD8 T cells ($n = 222$). The $p$ values were generated from the two-sided Wilcoxon rank-sum test.

distinctions between these two populations. Specifically, we observed the upregulation of CXCL13 in both PB-unrelated effector T cells and PB-unrelated memory T cells. CXCL13, a chemokine crucial for leukocyte trafficking and lymphoid tissue organization, has been implicated in T cell exhaustion. The upregulation of CXCL13 in PB-unrelated CD8 T cells may signify an indicative mechanism driving enhanced T cell exhaustion within these subsets (Dai et al, 2021; Yang et al, 2021). Additionally, we observed the downregulation of FGFBP2 and the upregulation of RPS26 in PB-unrelated effector T cells. FGFBP2, known for its involvement in regulating fibroblast growth factors (FGFs), negatively impacts T cells' survival, proliferation, and antigen responsiveness (Chang et al, 2014). Conversely, RPS26 acts as a checkpoint regulating T-cell survival and homeostasis. The upregulation of RPS26 could represent an adaptive survival response aimed at preserving cellular functions and homeostasis within the TME (Chen et al, 2021). Collectively, the coordinated upregulation of both CXCL13 and RPS26, along with the down-regulation of FGFBP2, may unveil a nuanced equilibrium between

exhaustion and survival within PB-unrelated CD8 T cell subsets. It implies how the TME influences immune cell behavior, leading to a trend of exhaustion and struggle for survival in T cells.

## TCRclub discovers the mechanisms underlying response to Anti-PD1 therapy in BCC patients

We conducted a study to analyze the response of T cells to therapeutic interventions in basal cell carcinoma (BCC) patients. The dataset used in this study comprises the paired scRNA-seq and scTCR-seq data of T cells obtained from site-matched tumors of BCC patients collected both before and after anti-PD1 therapy (Yost et al, 2019b; Data ref: Yost et al, 2019a). The dataset includes samples from a total of eleven patients, consisting of six responders and five non-responders to the therapy. We divided the dataset into four cohorts, responder-pre, responder-post, nonresponder-pre, and nonresponder-post, based on the patient's response to Anti-PD1 therapy and the time of sample collection (pre-treatment or post-treatment).

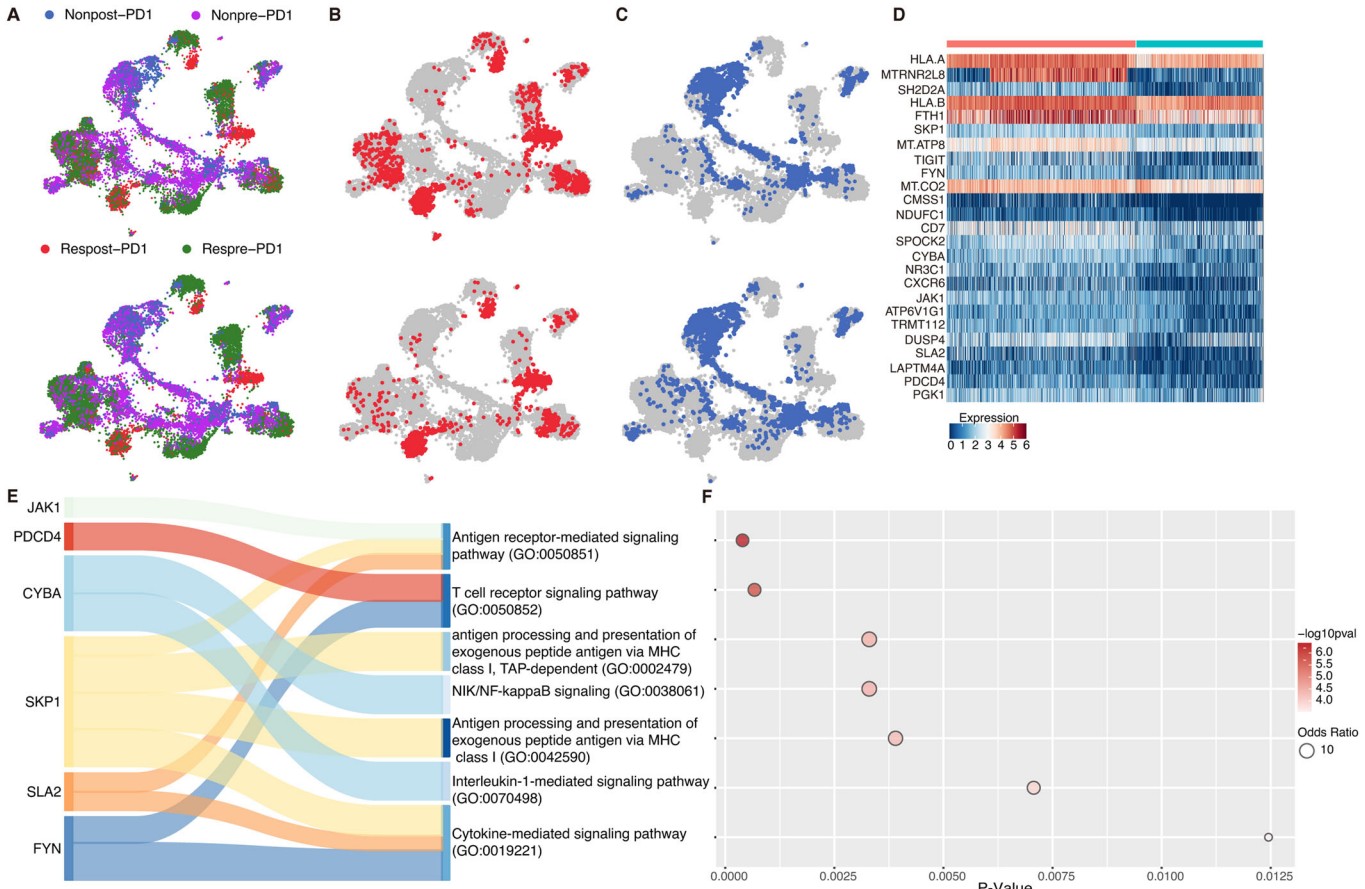

**Figure 4. Investigating the mechanisms of response to Anti-PD1 therapy in BCC patients using TCRclub.**

(A) UMAP plots of T cells colored according to the original cell-type labels. The upper plot displays the original labels prior to applying TCRclub, while the lower plot shows the re-assigned labels after the TCRclub application. (B) UMAP plots of T cells colored according to the cell-type labels before and after applying TCRclub. The upper plot depicts the original Respost-PD1 labels before the TCRclub application, while the lower plot highlights the re-assigned Respost-PD1 cells in red after the TCRclub application. (C) UMAP plots of T cells colored according to the cell-type labels before and after applying TCRclub. The upper plot shows the original Nonpost-PD1 labels before the TCRclub application, while the lower plot highlights the re-assigned NonPost-PD1 cells in red after the TCRclub application. (A–C) share the same legend. (D) Heatmap illustrating the DEGs discovered by analyzing the post-PD1 T cells related to the pre-PD1 T cells in responders. The top 25 genes with the highest p values are depicted. (E, F), Sankey plot (E) and Bubble plot (F) presenting the pathway enrichment analysis for the DEGs identified in the Respost-PD1 T cells related to the Respre-PD1 T cells (Respost-PD1 T cells, $n = 671$; Nonpost-PD1 T cells, $n = 450$). Seven enriched pathways, sorted based on the $-\log 10$ p values, along with their associated DEGs in (D), are displayed. The p values were generated from the two-sided Wilcoxon rank-sum test.

We explored the regression coefficient between pairwise expression distance and pairwise embedding distance calculated by TCRclub for the responder and nonresponder cohorts. Our analysis did not reveal a significant difference in these regression coefficients between the responder-post and nonresponder-post cohorts (Appendix Fig. S16). However, TCRclub offers an alternative perspective on the origins of T-cell clone proliferation, unlike the previous discovery that suggested the proliferation did not originate from pre-existing tumor-infiltrating T cells (Yost et al, 2019b). We defined four cell types based on the patient's response to therapy and whether the TCRs exclusively emerged in the post-treatment sample. For responders, we classified T cells as "Respost-PD1" cells if their CDR3$\beta$ sequences were only detected in the post-treatment sample. Otherwise, they were labeled as "Respre-PD1" cells. Similarly, the definitions "Nonpre-PD1" and "Nonpost-PD1" were used for the non-responders. We aimed to explore whether the post-PD1 cells were related to the pre-PD1

cells. Consequently, we re-assigned the cell-type label for each T-cell clone based on the majority voting results of its corresponding club. The clustering results were visualized using UMAP (Fig. 4A). Interestingly, we observed that in both responders and non-responders, a subset of T cells changed their labels due to clustering with another cell type. However, this change was more pronounced in responders (Fig. 4B,C). Specifically, a high proportion of T cells initially labeled as post-PD1 in responders were re-labeled as pre-PD1 cells, indicating that more post-PD1 cells were related to the pre-treatment batch in responders than non-responders.

We extracted the post-PD1 cells related to the pre-PD1 cells in two cohorts: responders and non-responders, and identified differentially expressed genes (DEGs) in responder vs. nonresponder (Fig. 4D; Appendix Fig. S17). Among the DEGs of responders, we obtained some clues concerning the mechanism of the response to Anti-PD1 therapy. For instance, HLA-A, a member of MHC

class I genes, plays a critical role in presenting antigens to cytotoxic T cells. When HLA-A is highly expressed, it increases the likelihood of presenting tumor-specific antigens to T cells, thereby enhancing T-cell recognition and response against tumor cells (Krensky, 1997). We also found JAK1 and SLA2 genes in the DEGs of responders. They are involved in mediating the signaling of various cytokines and growth factors that regulate the activation and differentiation of immune cells (Wu et al, 2023; Albacker et al, 2017). The identified DEGs highlight the involvement of specific immune-related pathways, including antigen presentation and immune signaling, contributing to an effective response to PD1 therapy among responders.

To further elucidate the functional implications of the DEGs in responders, we conducted pathway enrichment analysis (Appendix Fig. S18) and ranked the results based on their $-\log 10$ $p$ values. Seven pathways particularly related to immune response were selected (Fig. 4E,F). These pathways encompass critical aspects of immune signaling and antigen presentation. For instance, pathways such as GO:0002479 and GO:0042590 facilitate the generation and presentation of tumor antigens to T cells, the primary targets of PD1 therapy (Niu et al, 2021). The efficiency and quality of these pathways can significantly impact the recognition and elimination of tumor cells by the immune system. Noteworthy is the signaling pathway GO:0038061, which regulates the expression of genes involved in immune response, inflammation, cell survival, and apoptosis. This pathway may modulate the activity and regulation of NF-kappaB, a transcription factor controlling the expression of PD-L1 and other immune checkpoints (Yi et al, 2022).

## TCRclub characterizes antigen-specific T-cell response and gene patterns in COVID-19 patients

Understanding the strength and distribution of T-cell responses to SARS-CoV-2 is essential for elucidating the pathogenesis of COVID-19 and developing effective therapies. In this study, we analysed the paired scRNA-seq and scTCR-seq data from the dataset consisting of 41 peripheral blood mononuclear cell (PBMC) samples from COVID-19 patients with disease severity, including asymptomatic (AS) patients, moderate patients, severe patients, and severe recovery (SR) patients (Wang et al, 2022; Data ref: Zhang C, 2022). We used TCRanno (Luo et al, 2023) to predict the target antigens and virus for the T cells, and we adopted the club-based approach to determine the targeted antigens for each cell. T cells in a club were regarded as having the same antigen-specificity as this club's majority voting antigen-specificity. Similarly, the club was assumed to be specific to the voted antigen.

Previous studies (Chen and John Wherry, 2020; Wan et al, 2020; Liu et al, 2020) have reported a negative correlation between peripheral blood T lymphocyte counts and the severity of COVID-19. However, the extent of T cell response specific to SARS-CoV-2 among different patients remains to be clarified. We investigated the percentage of T cells specific to SARS-CoV-2 in relation to different severity levels. Specifically, we found a trend indicating that the symptomatic patients may exhibit a higher proportion of T cells specific to SARS-CoV-2 compared to those AS patients (Fig. 5A). We also found that the individuals aged over 60 remarkably showed a larger amount of T cells against SARS-CoV-2 (Fig. 5B). This observation suggests that patients with severe symptoms or older adults display an intensified and more pronounced immune response. It also implies that a higher count of T cells tailored to SARS-CoV-2 does not necessarily correlate with effective infection control or preventing severe symptoms. Multiple factors could contribute to this scenario, including a potential decline in overall immune function due to immunosenescence and the presence of cytokine storms (Del Valle et al, 2020; Nikolich-Zugich et al, 2020; Diao et al, 2020). Additionally, there was no such significant difference in the percentage of T cells targeted for SARS-CoV-2 between males and females or between long-term and short-term patients (Appendix Fig. S19). We also compared the regression coefficients obtained by TCRclub across those cohorts and observed that the symptomatic patients tend to have higher regression coefficients than the AS patients. Besides, individuals aged over 60 displayed higher regression coefficients than those aged below 60 (Appendix Fig. S20).

To explore whether patients with varying severity levels exhibit similar preferences for V and J gene usage towards the same antigen, we analyzed the frequency of different V or J gene usage across various antigens in these patient groups. Among the 11 antigens in the SARS-CoV-2 repository of TCRanno, specifically, we selected nine antigens that appeared in all severity levels. For each antigen, we calculated the Pearson correlation between different severity levels. We found the V-gene usage correlation coefficients to be generally low among different severity groups (Fig. 5C), suggesting that different TCR V-gene segments participate in the immunological response to the same antigen in patients with varying severity levels. Contrarily, we observed a relatively higher correlation of J gene usage across different severity levels for the same set of antigens (Fig. 5D). This indicates a more conserved pattern of J gene usage in response to the same antigen regardless of disease severity. The variability of V-gene usage and the relative conservation of J gene usage can be reflective of the differential roles of V and J segments in antigen recognition of COVID-19. While V genes, being more diverse, contribute to a broad scope of antigen recognition, the J genes, being less diverse, could be more specific to certain antigenic epitopes.

In particular, patients with different severity levels exhibited a high correlation in both V-gene and J gene usage, specifically for ORF1ab, compared to other antigens. We quantified the percentage of different antigen-specific T-cell clubs for SARS-CoV-2 in different severity levels, and found most T-cell clubs were specific to ORF1ab regardless of the severity (Appendix Fig. S21). We further visualized the most frequent V-J gene pairs for T cells specific to ORF1ab in different severity levels (Fig. 5E). Notably, compared to symptomatic patients (SM patients, including moderate, severe, and SR patients), AS and moderate patients displayed a distinct and unique tendency towards the usage of V-J gene pairs, such as TRBV7-2:TRBJ2-1, and TRBV20-1:TRBJ1-1. These unique V-J gene pairs could contribute to clearing the ORF1ab-bearing SRARS-CoV-2, explaining the milder symptoms observed in AS patients. Additionally, we observed a remarkable overlap within the SM group, and the SM group also shared TRBV9:TRBJ2-1 and TRBV11-2:TRBJ2-7 with AS patients in the most frequent V-J gene pairs. Certain V genes, including TRBV7-9, TRBV20-1, and TRBV27, were commonly used across different severity levels (Fig. 5E). However, in our further investigation of the most frequent V-J gene pairs for T cells specific to the surface glycoprotein (the antigen eliciting the second-highest T-cell club response in Appendix Fig. S21) and ORF7b (the antigen with fewer

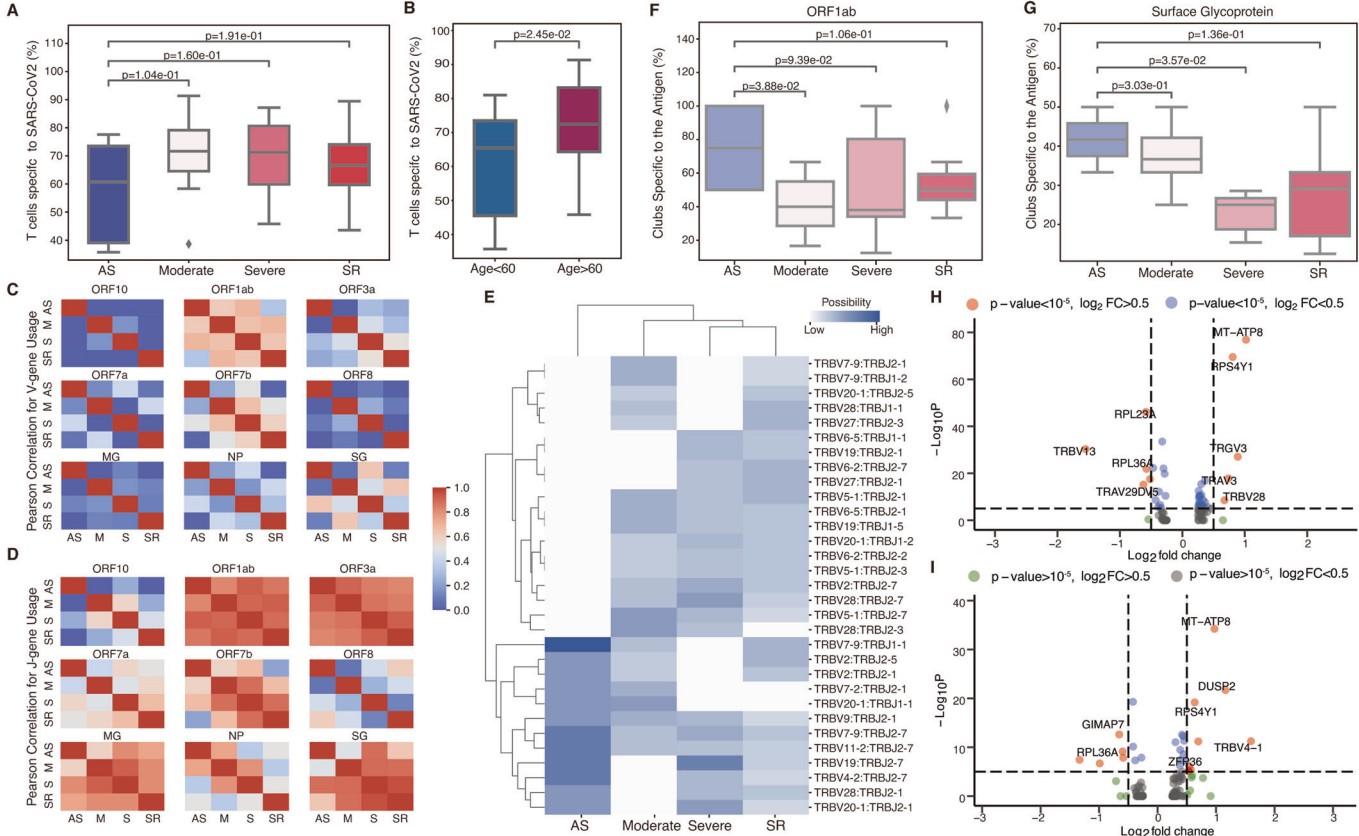

**Figure 5. TCRclub characterizes antigen-specific T-cell response and gene patterns in COVID-19 patients.**

(A, B) Percentage of T cells specific to SARS-CoV-2 across different severity levels (A) and age groups (B). Number of samples: AS, $n = 5$; Moderate, $n = 13$; Severe, $n = 11$; SR, $n = 12$; Age <60, $n = 20$; Age >60, $n = 21$. The p values were generated from the one-sided Mann–Whitney U-test. (C, D) Correlation of the usage of various V genes (C) and J genes (D) between different severity levels for different antigens. MG denotes membrane glycoprotein. NP denotes nucleocapsid phosphoprotein, and SG represents surface glycoprotein. (E) Most frequent V-J gene pairs for T cells specific to ORF1ab in different severity levels. (F, G) Percentage of T-cell clubs specific to ORF1ab (F) and surface glycoprotein (G) among patients with different severity levels. Number of samples: AS, $n = 4$; Moderate, $n = 11$; Severe, $n = 10$; SR, $n = 12$ (F) and AS, $n = 2$; Moderate, $n = 10$; Severe, $n = 6$; SR, $n = 10$ (G). The p values were generated from the one-sided Mann–Whitney U-test. (H, I) Volcano plots of DGEs for T-cell clubs specific to ORF1ab (H) and surface glycoprotein (I) in severe patients, compared with AS and moderate patients. The p values were generated from the two-sided Wilcoxon rank-sum test. Number of cells: AS and moderate, $n = 1291$; Severe, $n = 1318$ (H); AS and moderate, $n = 802$; Severe, $n = 457$ (I). Data information: For boxplots in (A, B, F, G), The central band represents the median. The lower and upper hinges represent the first and third quartiles, respectively. Whiskers are drawn to the farthest datapoint within ±1.5 × IQR from the nearest hinge.

T-cell clubs specific to, in Appendix Fig. S21), the overlap between the AS and SM groups was gradually inconspicuous (Appendix Fig. S22). The results suggest that T cells may have a relatively conserved V-J gene usage pattern to the most popular antigen, ORF1ab, that elicited the strongest T cell response in different severity levels.

We investigated the distribution of T-cell clubs specific to various antigens of SARS-CoV-2 across patients with different severity levels. We found that AS patients tend to have a higher number of clubs containing highly expanded T cells specific to ORF1ab or surface glycoprotein compared to SM patients (Fig. 5F,G, see Methods). Upon scrutinizing DEGs within T-cell clubs targeting ORF1ab and surface glycoprotein in severe patients compared to those in AS and moderate patients, we observed the upregulation of MT-ATP8 and the downregulation of RPL36A (Fig. 5H,I). MT-ATP8, known for its involvement in energy production and cellular stress, its upregulation in severe patients may signify heightened bioenergetic demands, mitochondrial

dysfunction, and pro-inflammatory activity in response to the same antigens (Asgari and Pousaz, 2021; Guarnieri et al, 2023). The downregulation of RPL36A, a key player in protein synthesis, suggests a potential impairment in the protein production capacity of T cells, which could adversely affect their growth and proliferation. Since ORF1ab and surface glycoprotein antigens are highly prevalent across different patients, these findings may imply an ineffective T-cell response to the predominant antigens in severe patients, thereby explaining the progression of severe symptoms.

## Discussion

Given the advancements in single-cell RNA and TCR sequencing technologies, there is a growing trend to integrate these data types for T-cell analysis across various research avenues, such as LRT (Xie et al, 2023), a tool for constructing cell trajectories, scNAT (Zhu et al, 2023) and mvTCR (Drost et al, 2024), tools for

representing data in a unified latent space. In this work, we presented the TCRclub, a novel approach for identifying functionally similar T cell groups. TCRclub integrates both TCR sequences and gene expressions to cluster T cells based on their local harmony. Local harmony refers to the nearby homogeneity among T cells that are functionally related and have similar characteristics (Uddin et al, 2022; Cover and Hart, 1967; Jaskowiak et al, 2011). Instead of generating flat cell groups after the integration, TCRclub considers the underlying nested structure of T cells and builds the cell hierarchy, resulting in a fine-grained resolution for capturing the functional diversity of T cells at the club layer.

We validated that TCRclub effectively captures the shared functional properties of T cells, achieving significantly higher purity (52.25%) and coverage (87.45%) than other methods in clustering T cells across over 400 pMHC categories. The remarkable clustering effectiveness (45.34%) suggests that TCRclub maintains a good balance between purity and coverage. A key feature of TCRclub is its model of the association between TCR distances and gene expression distances for identifying T cell clubs. Disregarding this association and relying solely on the T-cell representation obtained from the integration of scTCR-seq and scRNA-seq profiles may fall short of the identification of T-cell clubs. For example, applying the embeddings produced by scNAT (Zhu et al, 2023) and mvTCR (Drost et al, 2024) to identifying T-cell clubs resulted in lower clustering performance compared to TCRclub (Appendix Fig. S23). Additionally, we tested TCRclub's generalization performance on another dataset containing four samples and 44 pMHC categories from healthy individuals (Appendix Fig. S24). The higher average clustering effectiveness of TCRclub (77.24%) demonstrated its stable clustering performance on different datasets.

TCRclub proves to be a valuable tool with diverse applications. In CCA patients, ex vitro-expanded T cells have demonstrated feasibility for CCA treatment (Sawaisorn et al, 2024; Qiao et al, 2023). However, despite observing the successful migration of activated T cells from lymph nodes to the tumor primary site, we noted a limited population of activated T cells maintaining relations with lymph nodes or PB in tumor. Despite efforts to prolong their lifespan, such as upregulating the RPS26 gene, a substantial portion of activated T cells ultimately lose their external relations and transition into a late-stage state. Consequently, even if tumor-specific T cells can be expanded in vitro, favorable prognosis remains uncertain. Rapid exhaustion emerges as a barrier to prognosis due to the significant disturbances within the TME, underscoring the need to prioritize strategies aimed at enhancing the survival rate of T cells in such patients. Additionally, although TCR-related genes show different levels of expression in different types of CD8 cells, our observation that the majority of clubs contain different CD8 cell types in tumor samples suggests that as a tool considering both scRNA and TCR, TCRclub does not prioritize specific cell type genes over other genes. This indicates TCRclub's broader focus on capturing the overall functional landscape of T cells rather than concentrating solely on specific cell type markers.

PD1 therapy represents a cornerstone in cancer treatment, primarily by modulating the T cell repertoire and enhancing the activity of existing tumor-specific T cells (Donia et al, 2017; Gitto et al, 2021). In our investigation of pre-treatment and post-treatment samples from BCC patients using TCRclub, consistent with findings by Zhang et al (Zhang et al, 2021a), we observed a notable discrepancy between responders and non-responders. Specifically, non-responders exhibited a significantly smaller population of post-PD1 T cells correlating with pre-PD1 T cells. However, we contend that the inadequacy in T cell expansion alone may not entirely account for the unfavorable prognosis. Upon closer examination, we elucidated differences in the quality of post-PD1 T cells, derived from pre-PD1 T cells, between responders and non-responders. We observed responders displayed enhanced effectiveness in immune pathways related to immune signaling and antigen presentation. We postulate that responders possess a more conducive TME compared to non-responders, facilitating their ability to overcome immunosuppressive factors and exert potent anti-tumor activity under PD1 therapy. We advocate for future investigations leveraging more comprehensive TME samples, such as spatial transcriptomics data collected before and after treatment, to further substantiate this hypothesis.

Evidence from Zhao et al, (Zhao et al, 2017) suggests that T cells may be the core mediators for controlling the SARS-CoV virus. Previous studies (Sattler et al, 2020; Rümke et al, 2023) have suggested an overall impaired immune response in patients with COVID-19. Our study identified a more intense response of T cells to SARS-CoV-2 in severe patients, and this heightened response was also evident in the aged group. These results align with previous experiments (Peng et al, 2020; Moss, 2022; Sattler et al, 2020). Despite studies reporting strong pairing frequencies of VJ genes for B cells (Wen et al, 2020; He et al, 2021), there is a lack of attention given to VJ gene frequencies for T cells in COVID-19. We analyzed the frequency of VJ gene pairs of T cells for various antigens and symptoms. Our findings revealed distinct utilization patterns of V-J gene pairs in AS and moderate patients, such as TRBV7-2:TRBJ2-1 and TRBV20-1:TRBJ1-1, in relation to the most prevalent antigen, ORF1ab. However, further investigation is required to determine the effectiveness of these unique V-J gene pairs in clearing the ORF1ab-bearing SARS-CoV2. If their efficacy is confirmed, it could have implications for predicting the severity levels for COVID-19 patients. While some V-J gene pairs for ORF1ab were shared between AS and SM patients, there was a reduced shared preference for specific VJ gene pairs when targeting less prevalent antigens. Additionally, AS patients exhibited a higher incidence of T-cell clubs specific to prevalent SARS-CoV-2 antigens compared to severe patients. Our findings offer insights into the progression of severe symptoms and the potential development of targeted immunotherapies for SARS-CoV-2.

TCRclub offers unique insights when analyzing different datasets. In this study, we applied TCRclub to various disease cases and examined the distributions of the regression coefficients it generated. While significant differences in regression coefficients were observed among COVID-19 patients with varying symptoms or ages, no such differences were found between responders and non-responders among BCC patients, or across different tissues in CCA patients. However, when comparing different datasets, we found that they exhibited significantly different coefficients (Appendix Fig. S25). A higher regression coefficient indicates a steeper slope, meaning that changes in TCR diversity are accompanied by larger shifts in gene expression, suggesting a more dynamic immune response. This could reflect a more reactive or

adaptive immune environment, whereas lower coefficients may point to a more stable immune response. The variability of coefficients across different datasets underscores the importance of considering the unique molecular and cellular characteristics of each disease when analyzing T-cell behavior. Therefore, TCRclub holds the potential for facilitating cross-dataset studies.

In our current model, clubs are formed based on T-cell clones, with each clone assigned to only one club. However, the potential future assignment of a single clone to multiple clusters is considered, as it may provide deeper biological insights. Further refinement in T-cell processing would be beneficial, such as accounting for cells with multiple-TCR expressions or incorporating additional features like the expression of cell surface markers to more accurately capture the functional diversity of T cells. Our present approach primarily focuses on modeling the linear relationship between gene expression and TCR embeddings within a club to minimize intra-club distances. Future research could also consider inter-club distances to enhance understanding of T-cell interactions. Furthermore, we have extended the application of TCRclub to additional contexts (Appendix Supplementary Methods, Appendix Figs. S26–S28) and anticipate investigating broader applications of TCRclub in future studies.

Overall, the TCRclub approach incorporates gene expressions to interpret the TCR repertoire, effectively establishing a relationship between gene expressions and TCRs through local harmony. We expect TCRclub to contribute to developing immunotherapies and enhance our understanding of the immune system's response to various diseases.

# Methods

### Reagents and tools table

| Reagent/ Resource | Reference or source | Identifier or catalog number |
| --- | --- | --- |
| **Software** | | |
| Seurat v4.3.0 | https://github.com/satijalab/seurat/tree/seurat4 | |
| Scanpy v1.9.8 | https://github.com/scverse/scanpy | |
| Monocle v2.26.0 | https://github.com/cole-trapnell-lab/monocle-release | |
| TCRanno | https://github.com/deepomicslab/TCRanno | |
| SEAT | https://github.com/deepomicslab/SEAT | |
| Concept Saliency Maps | https://github.com/lenbrocki/concept-saliency-maps | |
| TCRclub | https://github.com/deepomicslab/TCRclub | |

### Overview of TCRclub

TCRclub integrates gene expressions and TCRs by modeling the agreement between pairwise TCR embedding distances and pairwise expression distances on the local harmony. Local harmony refers to the idea that neighbors are more likely to have similar characteristics and belong to the same category (Uddin et al, 2022; Cover and Hart, 1967). As illustrated in Fig. 1, after integrating gene expressions and TCR embeddings by local harmony, TCRclub builds the T-cell hierarchy based on the residual-distance matrix to identify T-cell clubs. Because the goal of TCRclub is to group functionally similar T cells into a club, and cells within the same clone (cells sharing the same CDR3$\beta$ sequences in a sample) are likely to have same antigen-specificity (Smith et al, 2021; Chronister et al, 2021) and tend to have similar gene expressions (Schattgen et al, 2022) (Appendix Fig. S1), we consider T-cell clones as single units instead of individual cells and groups the clones into clubs.

### Preprocessing for scRNA profile

In TCRclub, gene expression profiles are preprocessed by Scanpy (v1.9.8) before integration with the TCR embeddings. The scale factor for cell-level normalization is set to $10^4$ before applying natural logarithm transformation using log1p. Recognizing the informativeness of highly variable genes (HVGs), we incorporated HVGs in TCRclub. To determine the top ratio of HVGs used in TCRclub, we evaluated different ratios from the top 1% of HVGs to the top 60% of HVGs by measuring average clustering purity and clustering coverage on the dataset (Data ref: Francis et al, 2021) used in Fig. 2I. We observed a slight fluctuation in the clustering performance (Appendix Fig. S29). As increasing the proportion of HVGs did not necessarily improve the performance and could decrease clustering purity and coverage, we followed the approach of Zhang et al (Zhang et al, 2021b), retaining the top 10% of HVGs in our study. It is noted that the TCRclub users are not restricted to retaining only the top 10% of HVGs. The choice of input genes is user-friendly, allowing users to select input genes according to their specific needs.

### Embedding TCR sequences

We trained a neural network model with an encoder-classifier structure to generate discriminative embeddings, which are row vectors of 32 dimensions, for the CDR3$\beta$ amino acid sequences. The principal and structure design of the model has been described previously (Luo et al, 2023). We adapted the original model to fit our study. In essence, the network consists of a probabilistic encoder based on the Variation Autoencoder (Kingma and Welling, 2013) (VAE) and a classifier for the targeted epitope classes of TCRs. We selected TCR sequences with single-specificity (i.e., removing TCR sequences known to match multiple epitopes) from the IEDB databases (Vita et al, 2019), resulting in 107,042 unique TCR sequences against 627 epitopes in total. Further details can be found in Appendix Supplementary Methods and Appendix Fig. S30.

### Problem formulation

TCRclub receives matched gene expressions and TCR sequences (CDR3$\beta$ sequences) as input. Let $n$ denote the number of T-cell clones in the input. We use two symmetric matrices, $B \in \mathbb{R}^{n \times n}$ and $R \in \mathbb{R}^{n \times n}$, to represent the pairwise embedding distances and the pairwise expression distances between the clones, respectively. Here, $B_{ij}$ and $R_{ij}$ refer to the embedding distance and expression distance between clone $i$ and clone $j$, respectively. We investigated

the Pearson correlation between the pairwise TCR embedding distances and the pairwise gene expression distances for T-cell clones. Similar to the previous work (Zhang et al, 2021b), we observed a correlation for most of the datasets we studied, and the typical examples are shown in Appendix Fig. S31. The average Pearson correlation is 0.42 across all datasets (Appendix Table S1 and Appendix Fig. S32).

Based on the Pearson correlation between TCR embedding and expression distances that we observed, we assumed that there was a linear regression between the TCR embedding distances and expression distances within a club. Given a set of clones denoted as $\varphi_i$ which belong to the same club with clone $i$, we represent the regression coefficient for $\varphi_i$ as $a_i$. The optimization problem can be expressed as follows:

$$Minimize \sum_{j \in \varphi_i} \left( R_{ij} - a_i B_{ij} \right)^2. \qquad (1)$$

Let f denote the number of dimensions in TCR embedding. Let $\tau_i \in \mathbb{R}^{1 \times f}$ be the normalized TCR embedding of clone $i$, and $\tau_j \in \mathbb{R}^{1 \times f}$ be the embedding of clone j. We calculate $B_{ij}$ by the inner product $\tau_i \tau_j^t$, where $\tau_j^t$ is the transpose of $\tau_j$. A larger inner product value indicates higher proximity between the clones. To account for the relative importance of each dimension in the TCR embedding vector, we introduce a diagonal weight matrix $W \in \mathbb{R}^{f \times f}$ into the formulation, resulting in $\tau_i W \tau_j$. Let $T \in \mathbb{R}^{n \times f}$ denote the TCR embedding matrix of $n$ T-cell clones. Consequently, we can rewrite the pairwise embedding distance matrix B as $B = TWT^t$.

However, unlike the TCR embeddings, the RNA expression levels of T cells within a clone are not identical. Hence, we calculate the average expression distance between cells of clone $i$ and cells of clone $j$ as $R_{ij}$. Let g denote the number of genes. Assuming that the normalized expression vector for a cell $p$ from clone $i$ is denoted as $e_p \in \mathbb{R}^{1 \times g}$, and that the normalized expression vector for a cell $q$ from clone $j$ is denoted as $e_q \in \mathbb{R}^{1 \times g}$, we can calculate $R_{ij}$ as follows:

$$R_{ij} = \frac{\sum_{p \in i} \sum_{q \in j} e_p e_q}{\kappa_i \kappa_j}, \qquad (2)$$

where $\kappa_i$ and $\kappa_j$ refer to the number of T cells in clone $i$ and clone $j$, respectively.

Let $C \in R^{n \times n}$ be an indicator matrix containing binary values (0 or 1). The $i_{th}$ row vector of $C$ indicates the relationship between clone $i$ and other clones, where $C_{ij} = 1$ indicates that clone $j$ is clustered with clone $i$. We can reformulate the objective function for all the T clones in matrix form as follows:

$$Minimize \sum_{ij}^{n} C_{ij}(R - ATWT^t)_{ij}^2, \qquad (3)$$

where $A \in \mathbb{R}^{n \times n}$ is a diagonal matrix, with each clone $i$ assigned a customized coefficient $A_{ii}$ (i.e., $a_i$ in Eq. 1). The matrices $C$ and $A$ are used to define the relationship between clones and to provide customized coefficients for each clone through its linear regression, respectively. $W$, on the other hand, is a global parameter shared across all TCR embeddings. We include the L2-regularization term for $W$ in the objective function. This regularization ensures that some weights in $W$ do not become excessively large, which could

lead to potential overfitting to specific TCRs within the input. Specifically, we modify the objective function of TCRclub as follows:

$$Minimize \sum_{ij}^{n} C_{ij}(R - ATWT^t)_{ij}^2 + \beta \sum_{d=1}^{f} W_{dd}^2, \qquad (4)$$

where $\beta$ is a hyper-parameter that controls the strength of the regularization term.

### Ensure local harmony

TCRclub utilizes the concept of local harmony to estimate the correlation matrix $A$ and weights matrix $W$. Local harmony refers to the neighborhoods sharing similar characteristics and likely belonging to the same functional category (Uddin et al, 2022; Cover and Hart, 1967; Jaskowiak et al, 2011). We define $(R - ATWT^t)_{ij}^2$ as the residual-distance matrix. Based on the residual-distance matrix, clones with smaller residuals to a given clone, such as clone $i$, should be positioned closer to clone $i$ in the space. As a result, the neighboring clones, located in close proximity to clone $i$, are considered to share a similar function to the clone $i$.

In each iteration of TCRclub, we identify the nearest neighbors $k$ for each clone of T cells based on the residual-distance matrix and record the neighbors in the adjacent matrix $C$. If clone $j$ is one of the $k$ nearest neighbors of clone $i$, we set $C_{ij} = 1$; otherwise, $C_{ij} = 0$. The update of the correlation matrix $A$ and weights matrix $W$ occurs after the update of $C$. In other words, only the neighboring clones are considered when updating $A$ and $W$. By focusing on the local structure of the clones, this approach ensures that the estimation of $A$ and $W$ primarily relies on the local characteristics, enhancing our ability to capture the underlying biological signal and mitigating the risk of overfitting to noise. Once the parameters $A$ and $W$ are updated, we use the updated values to modify $C$, and this iterative process continues until the objective function converges.

### Parameter initialization

Here, we present the initialization procedure for the model parameters $A$, $W$, and $C$. We initialize the matrices $A$ and $W$ as the identity matrices to obtain a good starting point. However, initializing $C$ is more intricate as it involves identifying cell neighbors based on the initial residual-distance matrix. We employ a probabilistic sampling approach, as described in the Appendix Supplementary Methods, to initialize $C$. This approach serves as the default initialization method for $C$ in TCRclub. Specifically, we use the residual distances to calculate the likelihood of sampling $k$ neighbors for each T-cell clone. If clone $j$ has lower residual distances to clone $i$, it is more likely to be selected as one of the initial-nearest neighbors, and we set $C_{ij} = 1$ accordingly. Although a fixed initialization could be used (see Appendix Supplementary Methods), it may not provide an optimal starting point. A fixed initialization may yield the same suboptimal starting point every time, resulting in suboptimal outcomes (Gul et al, 2023). Using the probabilistic approach, we can sample different subsets of neighbors for each T-cell clone, resulting in a more diverse set of starting points (Shahriari et al, 2022; Lee and Stöger, 2023). This diversity increases the probability of identifying the optimal starting point, leading to better overall outcomes.

### Parameter inference

As the derivations for different parameters are similar, here, the derivation of the solution for $A$ is presented as an example. The complete inference can be found in Appendix Supplementary Methods. In TCRclub, the solutions for $A$ and $W$ are analytical. The solution for $A$ can be obtained by denoting the objective function (Eq. 4) as $L$, and then solving for $\frac{\partial L}{\partial A_{ii}} = 0$. This yields:

$$A_{ii} = \frac{\sum_j^n (TWT^tR)_{ij}}{\sum_j^n C_{ij}(TWT^t)_{ij}^2}. \tag{5}$$

By updating the values of $A$ and $W$ using these analytical solutions, we can improve the accuracy and robustness of the TCRclub algorithm.

### Build cell club hierarchy

Once the objective function (Eq. 4) has converged, the stable residual-distance matrix $\Delta$ is obtained. We employ our recent work, SEAT (Chen and Li, 2022), a tool for hierarchical clustering, to build a cell hierarchy at the level of clones based on $\Delta$. This hierarchy is a nested structure illustrating cellular functional diversity. The root layer represents the entire population, while the last layer corresponds to the smallest individuals (i.e., T-cell clones in our study). The penultimate layer, known as the club layer, represents the finest resolution of similarity in the SEAT algorithm, which best reflects functional similarity. TCRclub focuses on this club layer to identify clones with high functional similarity, as clustering performance significantly decreases when using other layers (Chen and Li, 2022).

We construct a sparse graph $G_s = (V, E_s)$ from $\Delta$, where $V$ denotes the set of T-cell clones and $E_s$ represents the binary edges between them. Based on $G_s$, we can form a tree-like structure denoted as $\Upsilon$, and the club hierarchies can be obtained by divisively finding the minimum structure entropy $S^{\Upsilon}(G_s)$. The entropy $S^{\Upsilon}(G_s)$ is calculated as the sum of the entropies of all the nodes in the tree and the entropies of all the T-cell clones, as defined by

$$S^{\Upsilon}(G_s) = \sum_{\mu \in \Upsilon} S^{\Upsilon}(G_s; \mu) + \sum_{u \in V} S^{\Upsilon}(G_s, u), \tag{6}$$

where $\mu$ represents the nodes of $\Upsilon$ and $u$ denotes the clones.

Initially, all the clones are coded in a root node $r$, and $r$ is also the leaf because $\Upsilon$ has 0 height. The leaf can be split into two children by maximizing the change in structure entropy $\delta_s(r)$, and the bipartition split is based on the Fielder vector (see Appendix Supplementary Methods). Clones with smaller Fielder values should be placed in the left child. The children can then be treated as the new leaves and repeatedly split until each leaf contains only two clones or the structure entropy change $\delta_s$ is less than a threshold value $\zeta$. When the divisive splitting ends, each leaf of $\Upsilon$ can be assumed as a T-cell clone club.

### Consensus results of TCRclub

In order to reduce the likelihood of obtaining suboptimal outcomes and enhance overall accuracy, we executed the workflow (Fig. 1A) $M$ times to obtain the consensus outcomes (Fig. 1B). The $M$ results are sorted based on the values of the objective function at convergence (Eq. 4). We selected the $m$ results with the smallest objective function and used them to construct a consensus matrix $O$. Element $O_{ij}$ represents the frequency rate of clones i and j appearing in the same club among the $m$ runs. The final hierarchical structure of the clones is determined based on the consensus matrix $O$. By employing the consensus matrix, we identified stable and reliable clubs while avoiding the inclusion of any spurious clubs that may arise due to the stochastic nature of $C$ initialization.

### Experiment setting

TCRs are encoded into the embeddings of 32 dimensions (row vectors). T cells expressing more than one TCRβ sequence were filtered out, as the detected multiple-TCR-expression T cells may result from sequencing errors (Lee et al, 2017; Ulbrich et al, 2022; Pai and Satpathy, 2021). The default setting of TCRclub includes a local nearest-neighbor parameter $k = 10$, and a L2-regularization parameter $\beta = 10^{-7}$. The algorithm is repeated $M = 50$ times, and from these runs, we select $m = 15$ results with the smallest objective function to obtain a consensus clustering result. The divisive splitting process ends when the change in structure entropy is below a threshold of $\zeta = 10^{-4}$ for a single result or $\zeta = 5 \times 10^{-4}$ for a consensus result. Details can be found in Appendix Supplementary Methods and Appendix Fig. S33. There is no newly generated dataset in this study, and all the data used within this study can be retrieved from public databases (Data ref: Azizi et al, 2018; Data ref: Borcherding et al, 2020; Data ref: Francis et al, 2021; Data ref: Shi et al, 2022a; Data ref: Yost et al, 2019a; Data ref: Zhang C 2022; Data ref: 10X Genomics Datasets CD8+T cells of Healthy Donor 1-4, 2019) (also see Appendix Supplementary Table S1).

### Analysis of latent representations of TCRs

In our work, we leveraged saliency maps to highlight the critical features captured by TCRclub in TCR amino acid sequences. Saliency maps help identify the key features that machine learning models focus on when making predictions. To generate these maps, we adopted concept vectors and concept scores proposed by Brocki et al (Brocki and Chung, 2019).

The embeddings of the T-cell clones belonging to the selected club $c$ are denoted as $Z^+$ and exhibit a specific antigen-specificity identified by TCRclub, while the remaining embeddings from other clubs are represented as $Z^-$. The concept vector $z_c$ for club $c$ is computed as the difference between the average embeddings of the clones within club $c$ and those outside of it, yielding:

$$z_c = \frac{1}{n^+} \sum Z^+ - \frac{1}{n^-} \sum Z^-. \tag{7}$$

Here, $n^+$ and $n^-$ denote the number of clones within club $c$ and the number of clones outside club $c$, respectively.

To calculate the concept score for the T-cell clones $i$ in club $c$, we take the dot product between the concept vector $z_c$ and the embedding $\tau_c^i$ of clone $i$. Subsequently, we utilize the Deepexplain (Ancona et al, 2017) to generate the saliency map, which attributes importance scores to input features based on the concept scores obtained.

### Analysis of performance in clustering T cells into pMHC-specificity clusters

We evaluated each sample's clustering purity and coverage based on the unique CDR3$\beta$ s. Let $\theta$ denote the number of clusters in a sample containing at least two unique CDR3$\beta$ s. We denote $\gamma_i$ as the maximum number of unique CDR3$\beta$ s targeting the same pMHC in cluster $i$. Furthermore, $n_i$ represents the size of unique CDR3$\beta$ s in the $i_{th}$ cluster, and $N_\omega$ denotes the size of unique CDR3$\beta$ s in sample $\omega$. The purity $v_\omega$ and clustering coverage $r_\omega$ of sample $\omega$ are defined as follows:

$$v_\omega = \frac{\sum_i^\theta \gamma_i}{\sum_i^\theta n_i},$$

$$r_\omega = \frac{\sum_i^\theta n_i}{N_\omega}. \tag{8}$$

When compared with other methods, we introduce a new metric called clustering effectiveness to provide a more comprehensive evaluation of clustering performance. The clustering effectiveness of sample $\omega$ is defined as $e_\omega = v_\omega \times r_\omega = \frac{\sum_i^\theta \gamma_i}{N_\omega}$. This metric integrates both clustering purity and coverage into a single measure, allowing for a balanced comparison of methods, particularly when comparing methods that may differ in how they handle the trade-off between clustering purity and coverage.

In Fig. 2I, we present the results obtained from TCRclub, TESSA, GIANA, GLIPH, ClusTCR, and CoNGA using their default settings. If a method produced no T-cell cluster in a sample, then its purity and the coverage on the sample were regarded as zero. Given the notably lower clustering coverage of the other state-of-the-art methods compared to TCRclub, we attempted to improve their performance by adjusting the particular parameters known to increase clustering coverage, as outlined in their respective code documentation. We modified the parameter values through the provided parameter interface. In cases where there was no clarification regarding the parameter's impact on clustering coverage or if the parameter interface was not provided, the parameter remained unchanged. Despite these efforts, TCRclub consistently exhibited better performance (Appendix Fig. S34). When comparing TCRclub with the other evaluated methods across each sample, we ensured that TCRclub's clustering coverage was not lower than that of the evaluated methods for a fair comparison, as lower clustering coverage often accompanies higher purity.

### Analysis of dynamic journey in cholangiocarcinoma patients

Before integrating the samples from tumor primary locus, PB, or lymph node, we applied preprocessing steps to filter out non-proliferative T cells by removing single TCR clonotypes. Following the annotation by Shi et al (Shi et al, 2022b), we annotated the T cells into different T-cell types. Subsequently, we integrated the remaining T cells using Seurat (v4.3.0). We employed TCRclub with default settings on the integrated data.

Based on the distribution (Appendix Fig. S35), we observed a significant clustering pattern where the majority of T cells within each tissue type (tumor, PB, and lymph node) grouped with cells from the same tissue type. Consequently, we categorized PB (or lymph) T cells as "tumor-unrelated cells" if their club comprised sorely PB (or lymph) T cells. Conversely, PB (or lymph) T cells

were labeled as 'tumor-related cells' if their club included both PB (or lymph) T cells and tumor T cells. Trajectory and pseudotime analysis were conducted using Monocle (v2.26.0). Analysis of DEGs was conducted using Seurat (v4.3.0).

### Analysis of response to Anti-PD1 therapy in BCC patients

To determine the origin of T cells, we integrated the pre- and post-treatment samples for each patient. Prior to integration, we applied preprocessing steps to filter out non-proliferative T cells by removing single TCR clonotypes. The remaining T cells were then integrated using Seurat (v4.3.0), except for patients su002 and su007, who were excluded due to a lack of either pre-treatment or post-treatment samples after removing the single TCR clonotypes. Subsequently, we applied TCRclub with default settings on the integrated data. We re-assigned the cell-type labels for each T clone based on the majority voting results of its club, and the vote is based on clones.

We isolated post-PD1 cells associated with pre-PD1 cells in both responders and non-responders. Subsequently, we performed a comparative analysis of these two groups of cells to identify DEGs. The DEGs identified in the responder group were utilized to investigate enriched pathways using Seurat (v4.3.0).

### Analysis of SARS-CoV-2 antigen-specific T cells in different severity levels for COVID-19

Single TCR clonotypes were removed to filter out non-proliferative T cells. Because of the design of TCRanno (Luo et al, 2023), we clustered all the T cells for each symptom by Scanpy (v1.9.8), retaining only those cells within clusters predominantly expressing the CD8 gene. Then, we recorded the targeted antigen and virus predicted by TCRanno (Luo et al, 2023) for each T cell. We applied TCRclub with default settings on the dataset (Wang et al, 2022). The consensus antigen and virus for each T-cell club were defined as the antigen and virus present in over 50% of T cells within a club. A club can have multiple kinds of antigens and viruses. All the T cells of the same club are supposed to be valid for the consensus antigen.

When examining the distribution of the antigen-specific T-cell clubs of SARS-CoV-2 for each patient, we specifically focused on clubs containing highly expanded T cells, as these clubs may represent the most prominent response to the antigen. We concentrated on clubs with a size greater than ten, as these larger clubs, although comprising only 25% of the total number of clubs, accounted for over 60% of the total T-cell population in the majority of samples, representing the highly expanded T-cell populations (Appendix Fig. S36). Let $n_i$ denote the number of T-cell clubs with a size greater than ten for sample $i$, and let $a_x^i$ denote the number of T-cell clubs specific to antigen $x$ among $n_i$ (if applicable). The ratio of T-cell clubs specific to antigen $x$ in this sample is calculated as $a_x^i / n_i$.

Analysis of DEGs was conducted using Seurat(v4.3.0).

## Data availability

The computer code produced in this study is available in the following database: Modeling computer scripts: Github (https://github.com/deepomicslab/TCRclub)

The source data of this paper are collected in the following database record: biostudies:S-SCDT-10_1038-S44320-024-00070-5.

## Peer review information

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

## Acknowledgements

We express our gratitude for the support provided by the National Key R&D·Program of China (Grant No. 2023YFC3403200). We appreciate Dr. Yuwei Zhang's valuable suggestions on the manuscript and Dr. Yiqi Jiang's valuable suggestions on the figures.

## Author contributions

**Yiping Zou**: Software; Formal analysis; Validation; Investigation; Methodology; Writing—original draft; Writing—review and editing. **Jiaqi Luo**: Software; Formal analysis; Writing—review and editing. **Lingxi Chen**: Software; Writing—review and editing. **Xueying Wang**: Visualization; Writing—review and editing. **Wei Liu**: Visualization. **Ruohan Wang**: Investigation. **Shuai Cheng Li**: Conceptualization; Supervision; Methodology; Writing—review and editing.

Source data underlying figure panels in this paper may have individual authorship assigned. Where available, figure panel/source data authorship is listed in the following database record: biostudies:S-SCDT-10_1038-S44320-024-00070-5.

## Disclosure and competing interests statement

The authors declare no competing interests.

