## [Peer Review File · Molecular Systems Biology]

Identifying T-cell clubs by embracing the local harmony between TCR and gene expressions

Yiping Zou, Jiaqi Luo, Lingxi Chen, Xueying Wang, Wei Liu, Ruohan Wang, and Shuai Cheng Li

Corresponding author(s): Shuai Cheng Li (shuaicli@cityu.edu.hk)

Review Timeline:

Submission Date:	14th May 24
Editorial Decision:	10th Jul 24
Revision Received:	3rd Sep 24
Editorial Decision:	27th Sep 24
Revision Received:	2nd Oct 24
Accepted:	15th Oct 24

Editor: Poonam Bheda

Transaction Report:

10th Jul 2024

Manuscript Number: MSB-2024-12415

Title: Identifying T cell clubs by embracing the local harmony between TCR and gene expressions

Dear Dr. Li,

Thank you again for submitting your work to Molecular Systems Biology. We have now heard back from the three reviewers who agreed to evaluate your study. As you will see below, the reviewers appreciate that the proposed approach addresses a timely topic. However, they raise a series of concerns, which we would ask you to address in a major revision.

In particular, Reviewer 2 finds novelty of the TCR club algorithm unclear, mainly because they expected benchmarking against additional TCR clustering methods. We appreciate that you have already benchmarked your method against 4 other previously published methods; however, given the referee's concern, we would suggest additional benchmarking against 'CoNGA' (Schattgen Nature Biotech 2021). Regarding the other method suggested by the reviewer 'mvTCR' (Drost Nature Communications 2024), although the preprint was posted prior to submission of the your manuscript it was only published afterwards, and therefore we would not strictly require benchmarking against this method. However it may be beneficial to users deciding which method to use if you did include some comparisons to highlight the advantages of your method.

All other issues raised would need to be satisfactorily addressed. Please let me know in case you would like to discuss in further detail any of the comments, I would be happy to schedule a call. When submitting your revised manuscript, please carefully review the instructions that follow below. We perform an initial quality control of all revised manuscripts before re-review; failure to include requested items will delay the evaluation of your revision.

We require:

- 1) A .docx formatted version of the manuscript text (including legends for main figures, EV figures and tables). Please make sure that the changes are highlighted to be clearly visible. Alternatively you may choose to submit your manuscript as a LaTeX file.
- 2) Individual production quality figure files as .eps, .tif, .jpg (one file per figure). For guidance, download the 'Figure Guide PDF' (<https://www.embopress.org/page/journal/17574684/authorguide#figureformat>).
- 3) At EMBO Press we ask authors to provide source data for the main figures. Our source data coordinator will contact you to discuss which figure panels we would need source data for and will also provide you with helpful tips on how to upload and organize the files.
- 4) A .docx formatted letter INCLUDING the reviewers' reports and your detailed point-by-point responses to their comments. As part of the EMBO Press transparent editorial process, the point-by-point response is part of the Peer Review File (PRF), which will be published alongside your paper.
- 5) A complete author checklist, which you can download from our author guidelines (<https://www.embopress.org/page/journal/17574684/authorguide#submissionofrevisions>). Please insert information in the checklist that is also reflected in the manuscript. The completed author checklist will also be part of the PRF.
- 6) Please note that all corresponding authors are required to supply an ORCID ID for their name upon submission of a revised manuscript.
- 7) It is mandatory to include a 'Data Availability' section after the Materials and Methods. Before submitting your revision, primary datasets produced in this study need to be deposited in an appropriate public database, and the accession numbers and database listed under 'Data Availability'. Please remember to provide a reviewer password if the datasets are not yet public (see <https://www.embopress.org/page/journal/17574684/authorguide#dataavailability>).

This study includes no data deposited in external repositories.

8) All Materials and Methods need to be described in the main text using our 'Structured Methods' format, which is required for all research articles. According to this format, the Methods section includes a Reagents and Tools Table (listing key reagents, experimental models, software and relevant equipment and including their sources and relevant identifiers) followed by a Methods and Protocols section describing the methods using a step-by-step protocol format. The aim is to facilitate adoption of the methodologies across labs. More information on how to adhere to this format as well as a downloadable template (.docx) for

the Reagents and Tools Table can be found in our author guidelines:
<https://www.embopress.org/page/journal/17444292/authorguide#structuredmethods>

An example of a Method paper with Structured Methods can be found here:
<https://www.embopress.org/doi/10.15252/msb.20178071>.

9) For data quantification: please specify the name of the statistical test used to generate error bars and P values, the number (n) of independent experiments (specify technical or biological replicates) underlying each data point and the test used to calculate p-values in each figure legend. The figure legends should contain a basic description of n, P and the test applied. Graphs must include a description of the bars and the error bars (s.d., s.e.m.). Please provide exact p values.

10) Our journal encourages inclusion of *data citations in the reference list* to directly cite datasets that were re-used and obtained from public databases. Data citations in the article text are distinct from normal bibliographical citations and should directly link to the database records from which the data can be accessed. In the main text, data citations are formatted as follows: "Data ref: Smith et al, 2001" or "Data ref: NCBI Sequence Read Archive PRJNA342805, 2017". In the Reference list, data citations must be labeled with "[DATASET]". A data reference must provide the database name, accession number/identifiers and a resolvable link to the landing page from which the data can be accessed at the end of the reference. Further instructions are available at .

11) We replaced Supplementary Information with Expanded View (EV) Figures and Tables that are collapsible/expandable online. A maximum of 5 EV Figures can be typeset. EV Figures should be cited as 'Figure EV1, Figure EV2" etc... in the text and their respective legends should be included in the main text after the legends of regular figures.

<https://www.embopress.org/page/journal/17574684/authorguide#expandedview>

13) Author contributions: CRedit has replaced the traditional author contributions section because it offers a systematic machine readable author contributions format that allows for more effective research assessment. Please remove the Authors Contributions from the manuscript and use the free text boxes beneath each contributing author's name in our system to add specific details on the author's contribution. More information is available in our guide to authors.

Please also suggest a striking image or visual abstract to illustrate your article as a PNG file 550 px wide x 300-600 px high. Share synopsis text and image, as well as eTOC:

Please note that these would be the final versions and changes during proofing are usually not allowed

16) As part of the EMBO Publications transparent editorial process initiative (see our policy here: https://www.embopress.org/transparent-process#Review_Process), Molecular Systems Biology will publish online a Peer Review File (PRF) to accompany accepted manuscripts.

In the event of acceptance, this file will be published in conjunction with your paper and will include the anonymous referee reports, your point-by-point response and all pertinent correspondence relating to the manuscript. Let us know whether you agree with the publication of the PRF and as here, if you want to remove or not any figures from it prior to publication. Please note that the Authors checklist will be published at the end of the PRF.

Molecular Systems Biology has a "scooping protection" policy, whereby similar findings that are published by others during review or revision are not a criterion for rejection. Should you decide to submit a revised version, I do ask that you get in touch after three months if you have not completed it, to update us on the status.

I look forward to receiving your revised manuscript.

Yours sincerely,

Poonam Bheda, PhD
Scientific Editor
Molecular Systems Biology

Reviewer #1:

This paper proposes TCR club, a novel computational tool for integrative analysis scTCR-seq and scRNA-seq. With the emergence of scTCR-seq and the increasing interest and importance, this work is timely. TCR club is based on a reasonable rationale and design, and its results seem to be promising. Especially, its benchmarking results (Fig 2f - 2j) are exciting because, as the authors also pointed out, existing algorithms (e.g., GLIPH, GIANA) often fail to make meaningful clusters. The purity and clustering coverage seem to be promising. However, I have some questions and comments for further clarification and confirmation of the results generated using TCR club. I provided more detailed comments below.

1. Section "Preprocessing for scRNA profile": The authors retained only the top 10% highly expressed genes. Why the top highly expressed genes, rather than highly variable genes? And why the top 10%? Can it be too strict? Are the results sensitive or robust to this choice? For example, if we use the top 1% or 50%, how will the results be changed?
2. Section "Problem formulation": Here the T cell clone is the unit, rather than each T cell, e.g., n is the number of T cell clones, and dimensions of B and R are $n \times n$. While this can be reasonable, it might not be straightforward to readers. Can the authors provide more justification and explanation for this?
3. Section "Problem formulation": In Equation (1), the authors assume the linear relationship (A) between R and B . I think this is not a trivial assumption. Can the authors provide justification for this choice?
4. Section "Problem formulation": In Equation (4), the L2 regularization is imposed on W , rather than A or C . I think this requires additional explanation. Can the authors provide more explanation about this L2 regularization?
5. Section "Analysis of SARS-CoV-2 antigen-specific T cells in different severity levels for COVID-19": Here the authors focused on T cell clubs, sizes of which are larger than ten. Can the authors provide justification of using ten as a cutoff? How many T cell clusters were retained through this filtering?
6. Section "TCRclub demonstrates better performance in clustering T cells into pMHC-specificity clusters": Here the results themselves are quite exciting. However, still, in general, the validation of TCR analysis results is not straightforward. Fortunately, the data they used were based on the experimental validation and had sufficient number of pMHC categories. How much can the results here be generalized? For example, the results here can still remain valid for other biological conditions?
7. Section "TCRclub demonstrates better performance in clustering T cells into pMHC-specificity clusters": It is exciting to see a tree-like structure for T cells in Figure 2f. Here, can the identified tree structure be further interpreted? For example, what if we cut the tree at one depth from the root, two depths from the root, etc.? Will the clusters identified in this way be different from each other in a biologically meaningful way?

Reviewer #2:

In this manuscript, the authors state to combine TCR and transcriptome information to cluster functionally related T cells/TCRs. While this topic is of interest, the authors don't relate to several relevant previous works. These are: <https://www.nature.com/articles/s41467-024-49806-9> (was out as preprint prior to peer-reviewed publication), <https://www.nature.com/articles/s41587-021-00989-2>. Furthermore, very related such as <https://pubmed.ncbi.nlm.nih.gov/34132766/> is also not cited. Therefore, it is unclear to what extent this work is conceptually

novel and provides an advance over the state of the art.

Reviewer #3:

In this study, TCRclub, a novel approach was developed to integrate scRNA-seq and scTCR-seq data to cluster functionally-similar T cells based on local harmony. Local harmony denotes the homogeneity of nearby T-cell neighbors, owing to the likelihood of similar characteristics and categorization. The performance of TCRclub was validated with a dataset with experimentally determined pMHC specificity. Application to cholangiocarcinoma and COVID patients generated interesting results. This is a valuable tool for coupled scRNA-seq and scTCR-seq data analysis. The manuscript can be improved if they can address or discuss the following issues.

- 1) The study provides a good tool to integrate coupled scRNA-seq and scTCR-seq data. α_i in equation (1) is an important parameter for integration. It is helpful to show the distributions of α . Are the distributions comparable across datasets and samples?
- 2) TCR-related genes show different levels of expression in different types of CD8 cells. Usually, their expression is higher in CD8 effector cells, and lower in the naïve cells. Does the different levels of expression affect the results?
- 3) Purity and coverage are used to assess the performance in this study. Is it possible to assess the performance with Adjusted Rand Index?
- 4) In both scRNA-seq and scTCR, there are many dropouts. How do dropouts affect the results? Or does this tool only use the cells with TCR?
- 5) There are many diseases with extreme clonal expansion of T cells. How does clonal expansion affect the performance of this tool?
- 6) There are private and public TCRs among different individuals, and scRNA-seq datasets have batch effects. It would be great if the authors could examine how these affect the result.
- 7) The same clones can be assigned to different TCRclub groups. This may be reasonable biologically, the author should discuss this in the manuscript.
- 8) It is natural to expect the agreement of scRNA-seq and scTCR-seq. I am curious whether the agreement differs in different samples/datasets.

Dear Dr. Bheda and Reviewers,

We feel great thanks for your professional review work on our manuscript, entitled "Identifying T-cell clubs by embracing the local harmony between TCR and gene expressions". Those comments are valuable and helpful for revising and improving our paper. We have studied the comments carefully, rerun the analysis and corrected any typos. Revised portions are highlighted on the paper. Please find our point-to-point responses below and my revisions/corrections in the re-submitted files. Thanks again!

Response to the comments of Reviewer #1:

This paper proposes TCR club, a novel computational tool for integrative analysis scTCR-seq and scRNA-seq. With the emergence of scTCR-seq and the increasing interest and importance, this work is timely. TCR club is based on a reasonable rationale and design, and its results seem to be promising. Especially, its benchmarking results (Fig 2f - 2j) are exciting because, as the authors also pointed out, existing algorithms (e.g., GLIPH, GIANA) often fail to make meaningful clusters. The purity and clustering coverage seem to be promising. However, I have some questions and comments for further clarification and confirmation of the results generated using TCR club. I provided more detailed comments below.

Response: Thank you for providing valuable suggestions that have greatly improved the quality of our manuscript.

[Comment 1]. Section "Preprocessing for scRNA profile": *The authors retained only the top 10% highly expressed genes. Why the top highly expressed genes, rather than highly variable genes? And why the top 10%? Can it be too strict? Are the results sensitive or robust to this choice? For example, if we use the top 1% or 50%, how will the results be changed?*

Response: Thank you for pointing this out. We feel sorry for our carelessness. We actually used highly variable genes (HVGs) in TCRclub. Thanks for your careful checks and we have corrected the typo in our resubmitted manuscript.

As shown in the figure above (also is the Appendix Figure S26), we investigated TCRclub's average clustering purity and clustering coverage on the dataset used in Fig 2l by adopting different top ratios of highly expressed genes (HVGs) from the top 1% of HVGs to the top 60% HVGs. We observed a slight fluctuation in the clustering performance (Appendix Figure S26). As increasing the proportion of HVGs did not necessarily improve the performance and could decrease clustering purity and coverage, we followed the approach of Zhang *et al.*, retaining the top 10% of HVGs in our study. We have added the explanation in Lines 393-400 on Page 12.

[Comment 2]. Section "Problem formulation": Here the T cell clone is the unit, rather than each T cell, e.g., n is the number of T cell clones, and dimensions of B and R are $n \times n$. While this can be reasonable, it might not be straightforward to readers. Can the authors provide more justification and explanation for this?

Response: Thanks for your suggestion and we apologize for any confusion caused. Because the goal of TCRclub is to group functionally similar T cells into a club, and cells within a clone are likely to have the same antigen-specificity [1,2] and tend to have similar gene expressions[3] (Appendix Figure S1), TCRclub considers the T-cell clone as the basic unit of analysis. Therefore, when TCRclub takes paired scRNA expression and TCR as input, it uses the T-cell clones of the input. Specifically, let n denote the number of T-cell clones. TCRclub constructs the matrix B and R with shape $n \times n$ to represent the pairwise embedding distances and the pairwise expression distances between the clones. We have added the explanation for taking T-cell clones as the unit in Lines 82-84 on page 2 and Lines 387-390 on Page 12 of the revised manuscript.

[1] Smith, N. P., Ruiter, B., Virkud, Y. V., Tu, A. A., Monian, B., Moon, J. J., Love, J. C., and Shreffler, W. G. (2021). Identification of antigen-specific tcr sequences based on biological and statistical enrichment in unselected individuals. *JCI insight*, 6(13).

[2] Chronister, W. D., Crinklaw, A., Mahajan, S., Vita, R., Kos,alog̃lu-Yalçın, Z., Yan, Z., Greenbaum, J. A., Jessen, L. E., Nielsen, M., Christley, S., et al. (2021b). Tcrmatch: predicting t-cell receptor specificity based on sequence similarity to previously characterized receptors. *Frontiers in immunology*, 12:640725.

[3] Schattgen, S. A., Guion, K., Crawford, J. C., Souquette, A., Barrio, A. M., Stubbington, M. J., Thomas, P. G., and Bradley, P. (2022). Integrating t cell receptor sequences and transcriptional profiles by clonotype neighbor graph analysis (conga). *Nature biotechnology*, 40(1):54–63.

[Comment 3]. Section "Problem formulation": In Equation (1), the authors assume the linear relationship (A) between R and B . I think this is not a trivial assumption. Can the authors provide justification for this choice?

Response: We appreciate your suggestion, which has allowed us to enhance the clarity and justification of our approach. We investigated the Pearson correlation between the pairwise TCR embedding distances B and the pairwise gene expression distances R for T-cell clones. The gene expression distance between any two T cells was calculated using the dot product of the normalized gene expression vectors. A larger dot product indicates greater similarity or a smaller distance. We calculated the gene expression distances of any two T-cell clones by averaging the gene expression distances within the T-cell clones. Similarly, the pairwise TCR embedding distance between any two of the T-cell clones was computed as the dot product of the normalized TCR embeddings (A description of the calculation of R and B can be found in caption of Appendix Figure S27, or Lines 422-434, Lines 428-433 on Page 12).

Similar to the finding of Zhang *et al.*, we observed a correlation between R and B for most of the samples we studied. The typical examples are shown above (also is the Appendix Figure S27). The average Pearson correlation is 0.42 across all datasets (Appendix Table S1, and Append Figure S28). Based on such Pearson correlation between gene expression and TCR

that we observed, we assumed that there was a linear regression between R and B , and then formulated our model.

We have reorganized the paragraphs in the “Problem Formulation” section to provide a more detailed explanation of the reason behind assuming a linear relationship between RNA and TCR, the explanation can be found in Lines 413-417 on Page 12 of the revised manuscript.

[Comment 4]. Section "Problem formulation": In Equation (4), the L2 regularization is imposed on W , rather than A or C . I think this requires additional explanation. Can the authors provide more explanation about this L2 regularization?

Response: Thanks for your suggestion. We appreciate the opportunity to clarify this point. In the manuscript, we write the formulation in the matrix form for simplicity. As C is an indicator matrix containing binary values with 0 or 1. $C_{ij} = 1$ indicates that clone j is clustered with clone i . Therefore, the equation of the clone i with any of its clustered clone j can be equivalent to: $R_{ij} - a_i(w_{11}\tau_{i1}\tau_{j1} + w_{22}\tau_{i2}\tau_{j2} + \dots + w_{ff}\tau_{if}\tau_{jf})$, where a_i is a customized linear regression coefficient for clone i . However, W is a global parameter shared across all TCR embeddings. We include the L2-regularization term for W in the objective function. This regularization ensures that some weights in W do not become excessively large, which could lead to potential overfitting to specific TCRs within the input. We have added the explanation to address your concerns and hope that it is now clearer. The explanation has been included in Lines 438-442 on Page 13.

β	0	default setting (1e-7)	1e-4
Clustering purity (%)	51.74	52.25	51.78
Clustering coverage (%)	86.97	87.54	87.2

Through a hyper-parameter β , we can control the strength of the regularization term. As shown above, we observed that the appropriate strength of the L2-regularization on W helps improve both the clustering purity and coverage, compared with none of the L2-regularization. Exploration of the hyper-parameter setting can be found in Appendix Supplementary Methods.

[Comment 5]. Section "Analysis of SARS-CoV-2 antigen-specific T cells in different severity levels for COVID-19": Here the authors focused on T cell clubs, sizes of which are larger than ten. Can the authors provide justification of using ten as a cutoff? How many T cell clusters were retained through this filtering?

Response: Thanks for your question. When examining the distribution of the antigen-specific T-cell clubs of SARS-CoV-2 across patients with different severity levels, we specifically focused on clubs containing highly expanded T cells, as these clubs may represent the most prominent response to the antigen. We concentrated on clubs with a size greater than ten,

as these larger clubs, although comprising only 25% of the total number of clubs (see Fig A above), accounted for over 60% of the total T-cell population in the majority of samples (see Fig B above), representing the highly expanded T-cell populations. We have supplemented the explanation in Lines 570-574 on Page 15.

[Comment 6]. Section "TCRclub demonstrates better performance in clustering T cells into pMHC-specificity clusters": Here the results themselves are quite exciting. However, still, in general, the validation of TCR analysis results is not straightforward. Fortunately, the data they used were based on the experimental validation and had sufficient number of pMHC categories. How much can the results here be generalized? For example, the results here can still remain valid for other biological conditions?

Response: To address the concern regarding the generalizability of our results, we evaluated TCRclub's clustering performance using the default settings on another dataset consisting of samples from four healthy donors, which included 44 verified pMHC categories.

Fig A shown below is Fig A of Appendix Figure S24. We compared TCRclub with the T-cell clustering by the average clustering effectiveness, purity and coverage of the dataset. Clustering effectiveness, a new metric introduced in the revised manuscript, is defined as the product of clustering purity and coverage to capture the trade-off between these two critical aspects of T-cell clustering performance because the purity is calculated based only on the clustered T-cell clones, and in most scenarios, tuning the clustering to improve purity typically reduces the coverage, and vice versa (Details of the metrics have been described in Lines 111-113 on Page 3, Lines 529-533 on Page 14).

As shown above, the results showed a higher average clustering effectiveness of 77.24% for TCRclub compared to other methods, demonstrating its stable performance on different datasets. We have discussed the generalization of TCRclub in Lines 325-327 on Page 10 of the revised manuscript.

[Comment 7]. Section "TCRclub demonstrates better performance in clustering T cells into pMHC-specificity clusters": It is exciting to see a tree-like structure for T cells in Figure 2f. Here, can the identified tree structure be further interpreted? For example, what if we cut the tree at one depth from the root, two depths from the root, etc.? Will the clusters identified in this way be different from each other in a biologically meaningful way?

Response: Thanks for your suggestion, and we apologize for any confusion caused. As shown below (also is the Appendix Figure S3), we used the bar chart to display the purity of T-cell clone clusters, and we identified the clusters at various depths. These clusters show different levels of functional similarity. The number of clusters and the functional similarity increases from the root to the leaves. TCRclub specifically focuses on the club layer (the last subfigure), which is positioned just before the last layer (the individuals) and represents the

finest resolution of functional similarity. We have revised the text on Page 3, Lines 116-119 to address your concerns. We have also included the descriptions of the hierarchy in Lines 480-485 on Page 13.

Response to the comments of Reviewer #2:

In this manuscript, the authors state to combine TCR and transcriptome information information to cluster functionally related T cells/TCRs. While this topic is of interest, the authors don't relate to several relevant previous works. These are: <https://www.nature.com/articles/s41467-024-49806-9> (was out as preprint prior to peer-reviewed publication), <https://www.nature.com/articles/s41587-021-00989-2>. Furthermore, very related such as <https://pubmed.ncbi.nlm.nih.gov/34132766/> is also not cited. Therefore, it is unclear to what extent this work is conceptually novel and provides an advance over the state of the art.

Response: Thanks for your valuable suggestions. In the revised manuscript, we have addressed the relevant works that you mentioned. We have incorporated CoNGA and ClusTCR into the T-cell clustering benchmarks in Fig 21. As shown on the left, TCRclub achieves an average clustering purity of 52.25% and a coverage of 87.45%, as well as an effectiveness of 45.34%. The clustering effectiveness, defined as the product of the clustering purity and coverage, is introduced in the revised manuscript to capture the trade-off between clustering purity and coverage, as in most scenarios, tuning the clustering to improve purity typically reduces coverage, and vice versa (Lines 111-113 on Page 3, Lines 529-533 on Page 14). We observed that TCRclub achieved significantly higher performance on the three metrics than other methods, indicating its comprehensive ability to detect functionally similar clusters across various samples. The description of the comparison has been included in Lines 123-130 on Page 4-5.

Regarding mvTCR, since it is primarily a tool for generating T-cell embeddings rather than clustering, we managed to assess its integration performance by using the embeddings it produced. After obtaining the embeddings for each T cell by mvTCR's default setting, we applied our clustering algorithm to identify T-cell clusters using pairwise distances calculated from the embeddings (with the default setting of TCRclub). As shown on the right, we also observed that TCRclub shows better performance than mvTCR and scNAT (another T-cell embedding tools) in clustering T cells into pMHC-specificity clusters. We have discussed the result as part of an extended study comparing the T-cell representation tools in Lines 322-325 on Page 10.

Response to the comments of Reviewer #3:

In this study, TCRclub, a novel approach was developed to integrate scRNA-seq and scTCR-seq data to cluster functionally-similar T cells based on local harmony. Local harmony denotes the homogeneity of nearby T-cell neighbors, owing to the likelihood of similar characteristics and categorization. The performance of TCRclub was validated with a dataset with experimentally determined pMHC specificity. Application to cholangiocarcinoma and COVID patients generated interesting results. This is a valuable tool for coupled scRNA-seq and scTCR-seq data analysis. The manuscript can be improved if they can address or discuss the following issues.

Response: Thank you sincerely for your invaluable feedback. Your comments have been instrumental in enhancing the quality of our manuscript.

[Comment 1]. *The study provides a good tool to integrate coupled scRNA-seq and scTCR-seq data. α_i in equation (1) is an important parameter for integration. It is helpful to show the distributions of α . Are the distributions comparable across datasets and samples?*

Response: Thanks for your suggestion. We have analysed the distribution of α for all disease cases as well as across different datasets. While significant differences in regression coefficients were observed among COVID-19 patients with varying symptoms or ages, no such differences were found between responders and non-responders among BCC patients, or across different tissues in CCA patients. However, when comparing different datasets, we found that they exhibited significantly different coefficients (Appendix Figure S25). We have discussed the observed patterns in Lines 142-146 on Page 5, Lines 217-219 on Page 8, Lines 268-270 on Page 9 and Lines 364-370 on Page 11.

[Comment 2]. *TCR-related genes show different levels of expression in different types of CD8 cells. Usually, their expression is higher in CD8 effector cells, and lower in the naïve cells. Does the different levels of expression affect the results?*

Response: To address this concern, we have supplemented an analysis in the analysis of the CD8 T cells within the tumor of the cholangiocarcinoma patients (Lines 182-185 on Pages 6). As mentioned in methods, single clonotypes were filtered, remaining effector, memory and exhausted CD8 T cells within tumor were identified based on previous annotations. We analysed the CD8 T-cell celltype distribution in the clubs. As shown above, we observed the majority of clubs contain a variety of CD8 celltypes, suggesting that although TCR-related genes show different levels of expression in different types of CD8 cells, as a tool considering both scRNA and TCR, TCRclub does not prioritize the TCR-related genes over other genes. Consequently, the TCR-related genes do not affect the results. We have discussed this point in Lines 335-338 on Page 10.

[Comment 3]. *Purity and coverage are used to assess the performance in this study. Is it possible to assess the performance with Adjusted Rand Index?*

Response: Thanks for your suggestion. Adjusted Rand Index is indeed a robust metric for evaluating clustering performance when the true number of clusters is known, and when the clustering methods generate the same number of clusters. However, ARI may not act as a

good assessment when the number of clusters far deviates from the ground truth cluster number, which is the current fact of T cell clustering. Current T cell clustering produces functionally similar T cell groups, which could be far more than the true number of antigen-specificity; this is why current research [1,2] uses clustering purity and coverage to measure the performance. However, we acknowledge the limitations of using clustering purity and clustering coverage separately. As purity is calculated based only on the clustered T-cell clones, this could lead to an unintuitive assessment because in most scenarios, tuning the clustering to improve purity typically reduces coverage, and vice versa. To address this, we introduced a new metric in the revised manuscript called "clustering effectiveness," which is the product of clustering purity and coverage. This metric provides a more comprehensive evaluation of both clustering purity and coverage, capturing the trade-off between these two aspects of T-cell clustering performance (Lines 111-113 on Page 3, Lines 529-533 on Page 14).

[1] Ze Zhang, Danyi Xiong, Xinlei Wang, Hongyu Liu, and Tao Wang. Mapping the functional landscape of t cell receptor repertoires by single-t cell transcriptomics. *Nature methods*, 18(1):92–99, 2021.
 [2] Huang Huang, Chunlin Wang, Florian Rubelt, Thomas J Scriba, and Mark M Davis. Analyzing the mycobacterium tuberculosis immune response by t-cell receptor clustering with gliph2 and genome-wide antigen screening. *Nature biotechnology*, 38(10):1194–1202, 2020.

[Comment 4]. *In both scRNA-seq and scTCR, there are many dropouts. How do dropouts affect the results? Or does this tool only use the cells with TCR?*

Response: Thanks for your question and we apologize for any confusion caused. TCRclub is designed to work with paired scRNA and scTCR data as inputs. Therefore, this tool only uses the cells with TCR.

To examine whether the dropouts would affect TCRclub’s performance or not, we introduced additional dropouts into the raw gene expression counts on the dataset with verified pMHCs used in Fig 21. As shown above, for example, “0.01” indicates that 0.01% of the non-zero entries of the raw count matrix were randomly set to zero. “None” indicates that no additional dropouts were added. We found that the increased dropouts could negatively impact both average clustering purity and clustering coverage, compared to the default setting with no added dropouts, as shown above. We have supplemented this analysis in the manuscript on Lines 131-134 on Page 5.

[Comment 5]. *There are many diseases with extreme clonal expansion of T cells. How does clonal expansion affect the performance of this tool?*

Response: Thanks for your suggestion. TCRclub assigns T-cell clones rather than individual cells to clubs. To estimate whether the extreme clonal expansion affects the performance of TCRclub, we have conducted a simulation of extreme clonal expansion. The expression profiles of expanded cells were simulated by adding Gaussian noise to the original clonotype’s expression (Appendix Supplementary Methods). As the figure shows below, we expanded the T-cell clone CASSEVTLGNYGYTF (A-B) and stimulated the gene expression

(C-D). We observed that this condition does not change the clustering results compared to scenarios without such expansion (E-F). We have included the analysis in Lines 134-137, Page 5.

[Comment 6]. *There are private and public TCRs among different individuals, and scRNA-seq datasets have batch effects. It would be great if the authors could examine how these affect the result.*

Response: Thanks for your suggestion. To address this concern, we have added an explanation regarding how TCRclub handles the batch effects and TCRs from different samples. We took the integrated PB-tumor samples and unintegrated PB-tumor samples from different patients as an instance. We applied TCRclub to the groups, respectively. We calculated the ratio of clubs containing T-cell clones from both PB and tumor (i.e., mixed clubs). As shown on the left (Figure is also the Appendix Figure S10), the results showed a significantly higher ratio of mixed clubs in the integrated samples compared to the unintegrated ones. This finding suggests that integrating gene expression data is crucial for the joint analysis of different samples; otherwise, TCRclub tends to cluster T cells with those from the same sample.

For the T cells sharing identical CDR3 β sequences within the same sample, we defined them as a T-cell clone, and cells within a given clone tend to exhibit similar gene expressions [1] (Appendix Figure S1). However, we found that cells sharing the same CDR3 β sequences from different samples (public TCRs) should not be regarded as a T-cell clone, as we observed the similarity in gene expression among T cells sharing the same CDR3 β sequences differed between cells from the same sample and those from different samples (see the right figure, which is also the Appendix Figure S11). Therefore, in our work, when applying TCRclub to T cells from different samples, gene expressions were integrated prior to TCRclub's analysis. Clones sharing identical CDR3 β sequences from different

samples were treated as distinct clones, ensuring that all T-cell clones from each sample could be analysed.

A description of the analysis can be found in Lines 148-164 on Page 5.

[1] Smith, N. P., Rutter, B., Virkud, Y. V., Tu, A. A., Monian, B., Moon, J. J., Love, J. C., and Shreffler, W. G. (2021). Identification of antigen-specific tcr sequences based on biological and statistical enrichment in unselected individuals. *JCI insight*, 6(13).

[Comment 7]. *The same clones can be assigned to different TCRclub groups. This may be reasonable biologically, the author should discuss this in the manuscript.*

Response: Thanks for your suggestion, and we apologize for the ambiguity we caused. In TCRclub, T-cell clones, rather than individual cells, are assigned to clubs, ensuring that each clone is placed in only one club. Moreover, the potential biological relevance of assigning the same T-cell clone to multiple clubs could reflect deeper functional relationships. We hope to explore this possibility in future versions of TCRclub. We have included a discussion on this topic in the revised manuscript, specifically in Lines 371-372 on Page 11.

[Comment 8]. *It is natural to expect the agreement of scRNA-seq and scTCR-seq. I am curious whether the agreement differs in different samples/datasets.*

Response: Thanks for your suggestion. We investigated the Pearson correlation between RNA and TCR for T-cell clones. Consistent with Zhang *et al.*, we observed a correlation for most of the datasets we studied. Typical examples are shown in the left figure (also is the Appendix Figure S27). As shown in the boxplot (Appendix Figure S28), we calculated the Pearson correlation for multiple datasets, and we observed that the Pearson correlation could be significantly different between several datasets (P values < 0.05 are marked with *). The average value of the Pearson correlation is 0.42 across all datasets. Description of the agreement of scRNA-seq and scTCR-seq have been included in Lines 413-417, Page 12.

We greatly appreciate these valuable suggestions for improving our manuscript. We have carefully revised the manuscript to address the issues raised, and we hope the updated version will meet the high standards of the *Journal of Molecular Systems Biology*.

Yours sincerely,

Shuai Cheng Li, PhD

Professor, Department of Computer Science

City University of Hong Kong

Email: shuaicli@cityu.edu.hk

27th Sep 2024

Manuscript Number: MSB-2024-12415R

Title: Identifying T-cell clubs by embracing the local harmony between TCR and gene expressions

Dear Dr. Li,

Thank you for the submission of your revised manuscript to Molecular Systems Biology. We have now received the enclosed reports from the referees that were asked to re-assess it. As you will see the reviewers are now globally supportive and I am pleased to inform you that we will be able to accept your manuscript pending the following final amendments:

- 1) Please check the "Author Checklist" carefully and complete all relevant questions. Currently information about which sections the information is available in for your 'yes' responses is missing.
- 2) Please combine the Code Availability into the Data availability section, formatted according to the example below:
"The computer code produced in this study is available in the following database:
- Modeling computer scripts: GitHub (<https://github.com/SysBioChalmers/GECKO/releases/tag/v1.0>)
- [data type]: [full name of the resource] [accession number/identifier] ([doi or URL or identifiers.org/DATABASE:ACCESSION])"
- 3) Please rename "Disclosure and competing interest statement" to "Disclosure and competing interests statement".
- 4) Author contributions: Please remove it from the manuscript and specify author contributions in our submission system. CRediT has replaced the traditional author contributions section because it offers a systematic machine-readable author contributions format that allows for more effective research assessment. You are encouraged to use the free text boxes beneath each contributing author's name to add specific details on the author's contribution. More information is available in our guide to authors:
<https://www.embopress.org/page/journal/17574684/authorguide#authorshipguidelines>
- 5) Please upload the Reagents and Tools table as a separate file choosing the file type "Reagent Table".
- 6) Please place individual sections of the manuscript in the following order: Title page - Abstract & Keywords - Introduction - Results - Discussion - Methods - Data Availability - Acknowledgements - Disclosure and Competing Interests Statement - References - Figure Legends - Expanded View Figure Legends.
- 7) For the figures and figure legends, please take care of the following:
 - Please make sure to update the callouts of all figures in the main manuscript text (currently figure callouts are missing for Appendix figures S32-S36).
 - Please note that the exact p values are not provided in the legends of figures 5b, f-g.
 - Please indicate the statistical test used for data analysis in the legends of figures 3k-l; 4f; 5h-i.
 - Please note that the box plots need to be defined in terms of minima, maxima, centre, bounds of box and whiskers, and percentile in the legends of figures 5a-b, f-g.
 - Please note that information related to n is missing in the legends of figures 3k-l; 5a-b, f-i.
- 8) Appendix file: Please ensure Appendix figures S32-S36 are included in the Table of Contents and called out in the main manuscript.
- 9) Funding: Please note that funding information should be given in the "Acknowledgements" section (not in its own separate section).
 - Please check your synopsis text and image before submission with your revised manuscript. Please be aware that in the proof stage minor corrections only are allowed (e.g., typos).
- 10) As part of the EMBO Publications transparent editorial process initiative (see our policy here: https://www.embopress.org/transparent-process#Review_Process), Molecular Systems Biology will publish online a Peer Review File (PRF) to accompany accepted manuscripts. This file will be published in conjunction with your paper and will include the anonymous referee reports, your point-by-point response and all pertinent correspondence relating to the manuscript. Let us know whether you agree with the publication of the PRF and as here, if you want to remove or not any figures from it prior to publication. Please note that the Authors checklist will be published at the end of the PRF.
- 11) Please provide a point-by-point letter INCLUDING my comments as well as the reviewer's reports and your detailed responses (as Word file).

I look forward to reading a new revised version of your manuscript as soon as possible.

Yours sincerely,

Poonam Bheda, PhD
Scientific Editor

Reviewer #1:

The authors addressed my previous comments nicely and I believe this paper is now publishable in Molecular Systems Biology.

Reviewer #2:

The authors have addressed all of my comments.

Reviewer #3:

Almost of all my concerns are addressed. There are two minor suggestions.

1) I acknowledge the regression coefficient (α_i in equation 1) was thoroughly examined. It would be great if the authors can discuss what the difference means, or which biological factors drive the difference.

2) The author clarified that the T cell clones, instead of cells are assigned to TCRclub groups. There may be more than one α/β TCR for one T cell. It would be interesting to examine/discuss the multiple TCR-expression T cells.

Dear Dr. Bheda and Reviewers,

We feel great thanks for your professional review work on our manuscript, entitled "Identifying T-cell clubs by embracing the local harmony between TCR and gene expressions". We have made careful modifications to the revised manuscript based on the comments. All changes made are highlighted in the revised manuscript so that they can be easily identified.

Here are our responses to all comments one by one.

Response to editorial comments

1) Please check the "Author Checklist" carefully and complete all relevant questions. Currently information about which sections the information is available in for your 'yes' responses is missing.

Done.

2) Please combine the Code Availability into the Data availability section, formatted according to the example below:

"The computer code produced in this study is available in the following database:

- Modeling computer scripts: GitHub (<https://github.com/SysBioChalmers/GECKO/releases/tag/v1.0>)
- [data type]: [full name of the resource] [accession number/identifier] ([doi or URL or identifiers.org/DATABASE:ACCESSION])"

Done.

3) Please rename "Disclosure and competing interest statement" to "Disclosure and competing interests statement".

Done.

4) Author contributions: Please remove it from the manuscript and specify author contributions in our submission system. CRediT has replaced the traditional author contributions section because it offers a systematic machine-readable author contributions format that allows for more effective research assessment. You are encouraged to use the free text boxes beneath each contributing author's name to add specific details on the author's contribution. More information is available in our guide to authors:

<https://www.embopress.org/page/journal/17574684/authorguide#authorshipguidelines>

Done.

5) Please upload the Reagents and Tools table as a separate file choosing the file type "Reagent Table".

Done.

6) Please place individual sections of the manuscript in the following order: Title page - Abstract & Keywords - Introduction - Results - Discussion - Methods - Data Availability - Acknowledgements - Disclosure and Competing Interests Statement - References - Figure Legends - Expanded View Figure Legends.

Done.

7) For the figures and figure legends, please take care of the following:

- Please make sure to update the callouts of all figures in the main manuscript text (currently figure callouts are missing for Appendix figures S32-S36).

Done.

- Please note that the exact p values are not provided in the legends of figures 5b, f-g.

Done. We directly update the exact p values on the figures 5b, f-g.

- Please indicate the statistical test used for data analysis in the legends of figures 3k-l; 4f; 5h-i.

Done.

- Please note that the box plots need to be defined in terms of minima, maxima, centre, bounds of box and whiskers, and percentile in the legends of figures 5a-b, f-g.

Done.

- Please note that information related to n is missing in the legends of figures 3k-l; 5a-b, f-i.

Done.

8) Appendix file: Please ensure Appendix figures S32-S36 are included in the Table of Contents and called out in the main manuscript.

Done.

9) Funding: Please note that funding information should be given in the "Acknowledgements" section (not in its own separate section).

Done.

10) As part of the EMBO Publications transparent editorial process initiative (see our policy here: https://www.embopress.org/transparent-process#Review_Process), Molecular Systems Biology will publish online a Peer Review File (PRF) to accompany accepted manuscripts. This file will be published in conjunction with your paper and will include the anonymous referee reports, your point-by-point response and all pertinent correspondence relating to the manuscript. Let us know whether you agree with the publication of the PRF and as here, if you want to remove or not any figures from it prior to publication. Please note that the Authors checklist will be published at the end of the PRF.

Agree.

11) Please provide a point-by-point letter INCLUDING my comments as well as the reviewer's reports and your detailed responses (as Word file).

Done.

Point-by-point response to reviewers' comments.

Reviewer #1:

The authors addressed my previous comments nicely and I believe this paper is now publishable in Molecular Systems Biology.

Response: We thank the reviewer for finding our revised manuscript satisfactory.

Reviewer #2:

The authors have addressed all of my comments.

Response: We thank the reviewer for finding our revised manuscript satisfactory.

Reviewer #3:

Almost of all my concerns are addressed. There are two minor suggestions.

Response: We thank the reviewer for finding our revised manuscript satisfactory and for the valuable suggestions.

1) I acknowledge the regression coefficient (α_i in equation 1) was thoroughly examined. It would be great if the authors can discuss what the difference means, or which biological factors drive the difference.

Response: Thanks for your suggestion. We have discussed the regression coefficient further about the difference in Lines 367-370: "A higher regression coefficient indicates a steeper slope, meaning that changes in TCR diversity are accompanied by larger shifts in gene expression, suggesting a more dynamic immune response. This could reflect a more reactive or adaptive immune environment, whereas lower coefficients may point to a more stable immune response."

2) The author clarified that the T cell clones, instead of cells are assigned to TCRclub groups. There may be more than one α/β TCR for one T cell. It would be interesting to examine/discuss the multiple TCR-expression T cells.

Response: Thanks for your suggestion. In this study, T cells expressing more than one TCR β sequence were filtered out as the detected multiple-TCR-expression T cells may result from sequencing errors (Lee et al., 2017; Ulbrich et al., 2022; Pai and Satpathy, 2021) (Line 508). However, with the development of sequencing technology, we agree that in future work, further refinement in T-cell processing would be beneficial, such as accounting for cells with multiple TCR expressions, to more accurately capture the functional diversity of T cells (Line 375).

Lee, E. S., Thomas, P. G., Mold, J. E., & Yates, A. J. (2017). Identifying T cell receptors from high-throughput sequencing: dealing with promiscuity in TCR α and TCR β pairing. *PLoS computational biology*, 13(1), e1005313.

Ulbrich, J., Lopez-Salmeron, V., & Gerrard, I. (2022). BD rhapsody™ single-cell analysis system workflow: From sample to multimodal single-cell sequencing data. In *Single Cell Transcriptomics: Methods and Protocols* (pp. 29-56). New York, NY: Springer US.

Pai, J. A., & Satpathy, A. T. (2021). High-throughput and single-cell T cell receptor sequencing technologies. *Nature methods*, 18(8), 881-892.

We appreciate Dr. Bheda and the reviewers' warm work earnestly and hope that the correction will meet with approval.

Once again, thank you very much for your comments and suggestions.

Yours sincerely,

Shuai Cheng Li, PhD

Professor, Department of Computer Science

City University of Hong Kong

Email: shuaicli@cityu.edu.hk

15th Oct 2024

Manuscript number: MSB-2024-12415RR

Title: Identifying T-cell clubs by embracing the local harmony between TCR and gene expressions

Dear Dr. Li,

Congratulations on an excellent manuscript, I am pleased to inform you that your manuscript has been accepted for publication in the Molecular Systems Biology. Thank you for your comprehensive response to referee concerns. It has been a pleasure to work with you to get this to the acceptance stage.

Yours sincerely,

Poonam Bheda, PhD
Scientific Editor
Molecular Systems Biology
